# Neural implementation of computational mechanisms underlying the continuous trade-off between cooperation and competition

M. A. Pisauro [1,2,3,5] ✉, E. F. Fouragnan [1,3,4,5], D. H. Arabadzhiyska [3], M. A. J. Apps [1,2,6] & M. G. Philiastides [3,6]

Social interactions evolve continuously. Sometimes we cooperate, sometimes we compete, while at other times we strategically position ourselves somewhere in between to account for the ever-changing social contexts around us. Research on social interactions often focuses on a binary dichotomy between competition and cooperation, ignoring people's evolving shifts along a continuum. Here, we develop an economic game – the Space Dilemma – where two players change their degree of cooperativeness over time in cooperative and competitive contexts. Using computational modelling we show how social contexts bias choices and characterise how inferences about others' intentions modulate cooperativeness. Consistent with the modelling predictions, brain regions previously linked to social cognition, including the temporo-parietal junction, dorso-medial prefrontal cortex and the anterior cingulate gyrus, encode social prediction errors and context-dependent signals, correlating with shifts along a cooperation-competition continuum. These results provide a comprehensive account of the computational and neural mechanisms underlying the continuous trade-off between cooperation and competition.

In social interactions many species, including humans, often behave competitively – acts aimed at obtaining a resource at the expense of another benefitting—or cooperatively—acts aimed to benefit both self and other. Although people and animals commonly alternate between cooperative and competitive behaviours for access to resources, territories, and status[1–7] we are still lacking an integrated understanding of how the brain controls and arbitrates over the continuous trade-off between cooperation and competition, and more specifically, which neural mechanisms and computational principles are involved.

Classically, cooperation and competition have been treated as alternative social orientations, whereby one acts either cooperatively or competitively at any point in time[8]. In both one-shot and multi-rounds economic games, such predispositions are typically measured with social dilemmas requiring binary choices where people either cooperate or compete with a partner[9–11]. Yet, in the real world, behaviour is not so dichotomised. The common descriptions of people as being "fully cooperative" or "highly competitive" highlight that these behaviours are considered along a spectrum, and what may matter for social behaviour is one's degree of cooperativeness or competitiveness. But, how do people decide upon their degree of cooperation or competition? And how do they adjust it over time?

[1]Department of Experimental Psychology, University of Oxford, Oxford, UK. [2]Centre for Human Brain Health, School of Psychology, University of Birmingham, Birmingham, UK. [3]School of Psychology and Neuroscience, University of Glasgow, Glasgow, UK. [4]Brain Research Imaging Center and School of Psychology, Faculty of Health, University of Plymouth, Plymouth, UK. [5]These authors contributed equally: M. A. Pisauro, E. F. Fouragnan. [6]These authors jointly supervised this work: M. A. J Apps, M. G. Philiastides. ✉e-mail: m.a.pisauro@bham.ac.uk

Broadly speaking, previous research using games with binary choices suggests that cooperativeness is shaped by three factors: (i) the environment, where the availability of resources and their distribution shape choices[12], encouraging cooperation in rich and fair environments[13] while favouring competition when resources are scarce and unevenly distributed[14,15], (ii) personal predispositions and inherent social biases shaped by psychological traits[16–18] and (iii) how dyads interact with each other, with cooperation favoured by reciprocity[19] and the evolution of trust in repeated interactions[20,21] and the spread of reputational information within groups[22–24].

In economic games, environments are manipulated by the "payoff matrices" where changing the rewards available to each member of a dyad within an interaction influences behaviour, with people making more choices to act competitively or cooperatively when the payoff matrices favour it[25]. However, although economic theories assume people will eventually settle upon an optimal equilibrium, this is not always the case[26–28]. People have tendencies and psychological traits that lead to biases towards being more cooperative or more competitive in general, regardless of the payoff matrix.

Moreover, people's behaviour is determined by the psychological processes engaged when monitoring the behaviours of others. We monitor others' behaviour, and use mentalizing processes to infer their intentions, and adapt our cooperativeness accordingly[29]. At the core of this mechanism is the rewarding property of reciprocity in repeated interactions[19,30], which emerges through social learning driven by social prediction error signals[31]. However, to date, a formal account that unifies these features together and thus predicts someone's degree of competitiveness has not been forthcoming.

Research is increasingly showing that people's biases in social behaviour, and continuously updating inferences about others, can be captured by computational models, including those based on Bayesian principles[32–36]. In such accounts, model parameters can capture biases and people's expectations of others' behaviours which are updated by prediction errors (the surprise associated with the discrepancy between a prediction about another's action and their actual behaviour). Such Bayesian models have captured how people respond to the changing trustworthiness of other's advice and to behaviour in iterative economic games where people make binary choices[32,37]. Here, we propose to use Bayesian models to account for how people move along a cooperation-competition continuum based on their expectations of reciprocity of the co-player, their inherent social bias, and the incentives of the social environment.

Strikingly, regions of the brain that have been implicated in representing cooperative and competitive behaviours have been shown to do so by processing social prediction errors, that lead to an update in whether people behave cooperatively or competitively. In particular, portions of the temporo-parietal junction (TPJ), medial prefrontal frontal cortex (mPFC), anterior cingulate gyrus (ACCg), and portions of the anterior cingulate and paracingulate sulci corresponding to areas 24, 32 and 8 are all engaged when processing the competitive or cooperative behaviours of others[38–42]. The same regions have also been shown to signal prediction errors when monitoring others' behaviours, and in tasks requiring inferences to be made about the actions of others[32,37,43–47]. However, how these regions process information about the social context and use Bayesian signals relating to the cooperativeness of others to influence one's own degree of cooperation is poorly understood.

To test the notion that people behave on a continuum between cooperation and competition, we designed a new social game called the *Space Dilemma*. This game capitalises on a well-known economic principle controlling the spatial location of competitors in duopoly[48] and generalises to a continuum the trade-off between cooperation and competition which is dichotomised in the *Prisoner's Dilemma*[49,50]. In the game, two players decide whether and how much to compete or cooperate with each other, by positioning themselves in different locations of a continuous space, whereby each location is rewarded differently on a trial-by-trial basis. These decisions take place over multiple trials in three blocks with payoff matrices creating different social contexts that encouraged different degrees of cooperation and competition: (i) cooperative—where both players receive an equal amount of the reward, irrespective of who is best positioned (ii) competitive – where the best positioned player wins a reward while the other player incurs in a proportional loss and (ii) intermediate—where one player receives the reward and the other receives nothing. In each of these conditions, the best strategy would be to cooperate but with different, increasing risks associated with the defection of the co-player. Thus, to maximise rewards, players must consider what the optimal location is, but also infer the intentions of the other player, predict their level of cooperativeness and adapt one's location accordingly.

Here, we tested 27 pairs of participants playing the Space Dilemma whilst one in each pair underwent fMRI. We predicted that people would adapt their locations according to a general bias in cooperativeness, a shift in competitiveness across social contexts, but also trial-to-trial shifts in cooperativeness depending on the actions of the other player. We hypothesised that sub-regions of the TPJ and mPFC linked to processing information about others would signal (i) the degree of bias one has across the social contexts, (ii) prediction errors relating to the surprise associated with the other player's competitiveness and (iii) signal the degree to which one is updating one's behaviour due to the other player's competitiveness.

We show that people's behaviours are best predicted quantitatively by a Bayesian learner informed about the risk of losing and winning in each context, which also constantly updated behaviour based on the actions of the other player. We show that surprise signals are coded within clusters in the TPJ, in an unsigned manner in the posterior TPJ, but in a signed manner that correlated with updating subsequent behaviour in the anterior TPJ. In addition, distinct regions in mPFC, the ACCg and in the paracingulate sulcus carried information about participants' increases in cooperativeness and the degree to which they used trial-by-trial information about the other player, suggesting important roles in shifting behaviour along the continuum away from the default behaviour induced by the social context. These results provide a comprehensive characterisation of how the brain monitors and controls the continuous trade-off between cooperation and competition.

## Results

### The space dilemma

Pairs of participants, one inside the fMRI scanner and one in an adjacent room, played the game. All participants were told to imagine that they were foraging for food in a territory and were asked to make a prediction about the position of the food in a linear space (a straight line that represents the territory, Fig. 1a left panel). They were told that the target "food" would appear somewhere in the territory as its position was randomly sampled from a uniform distribution. They were then presented with a bar moving across the space (representing their location) and were required to commit to a certain location by pressing a button while the bar was moving in the linear space. This location would signal their prediction about the target position. Each player made her/his predictions and watched the other player's response. After the two players responded, the target appeared. On any trial, the participant who made the best prediction (closer to the target) won and got a reward which depended on the distance to the target: the lower the distance, the higher the reward (Fig. 1a).

As the target location is uniformly distributed across the space, if only one player would play the game, the optimal location to minimise the distance from the target and therefore maximise the average reward is the midpoint (Fig. 1b top panel, supplementary Fig. 1). With two players, the average total reward is maximised when the players

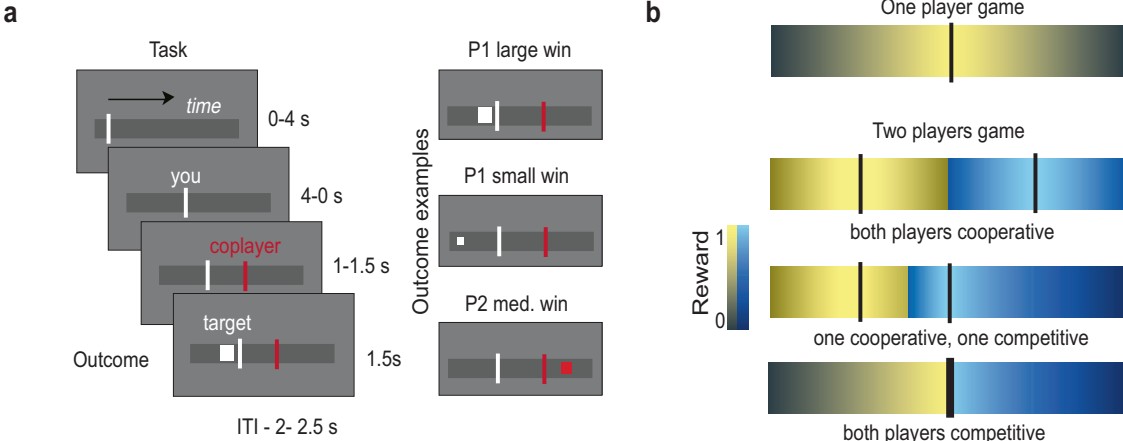

**Fig. 1 | Schematic representation of the Space Dilemma. a** Participants first positioned themselves in the space, hidden from the other player. They were then presented with a bar moving across the space (representing their location) and were required to commit to a certain location by pressing a button while the bar was moving through it. The bar would take 4 s to reach the end of the space. Once they responded, the bar stopped at the chosen location and was shown for the remainder of the 4 s. After both counterparts positioned themselves, their respective positions were shown to each other for 1–1.5 s before the target appeared (left panel). The player closer to the target won the trial (three examples in right panel) as identified by the colour of the target. The reward obtained is inversely proportional to the distance to the target, and reflected by the size of the target square. **b** The average reward for each player depends on the position in the territory. In each panel, the colour intensity represent the average reward obtained playing that position over many trials. In individual settings (top panel), the best strategy–to minimize distance to the target and maximize rewards - is to target the middle of the space. However, in the two-player space dilemma, as deployed here, multiple configurations exist. Fully cooperative behaviour involves both players positioning themselves in the midpoint of each hemifield, which minimizes the average proximity to any possible location of the target, thus maximising gains (second panel from the top). As this strategy is not a Nash equilibrium, players may have the incentive to deviate from their half side and thus cover more territory (third panel from the top). As such, any positioning closer to the midpoint can be defined as more competitive behaviour. When both players are highly competitive they both target the midpoint, winning less on average (bottom panel).

cooperate, by occupying the mid points of the two hemifields (Fig. 1b second panel, supplementary Fig. 1). However, one player might be tempted to occupy the midpoint, as this would maximise their own personal expected reward, at the expense of the other. As such, the closer a player gets to the midpoint, the more competitive his or her behaviour (i.e., less reciprocity towards the co-player's cooperation, Fig. 1b third panel). Crucially, this competitive behaviour would lower the total reward over all trials because when the target falls within that player's hemifield he/she has a higher probability of being further from it, thereby earning a smaller reward. Similarly, if both players choose to compete by trying to maximise their individual chance of winning going for the midpoint, they would expect to obtain the same reward, albeit reduced compared to the optimal locations when cooperating (Fig. 1b fourth panel).

We manipulated the social context by controlling the reward distributions (as determined by the $\alpha$ parameter, see Methods and Fig. 2a). We defined a cooperative context as one where participants shared the reward irrespective of the winner ($\alpha = 0.5$, Fig. 2a), and a competitive context in which losing a trial is associated with an economic loss whilst the winner sees its reward boosted by the same amount ($\alpha = 2$; Fig. 2a). An intermediate context was defined as one where the winner takes all the reward, while the loser in each trial did not receive neither a benefit nor a loss ($\alpha = 1$; Fig. 2a). To behave adaptively in the task, participants had to change their strategy according to both the co-player response and the social context.

Whilst the best long-term strategy to increase the total reward for the dyad in each context, unknown to the participants, was to always cooperate (supplementary material and Supplementary Fig. 1), this was not always the optimal strategy for individual players (which also depends on the co-player choices and is susceptible to end-game effects as the number of trials is finite) and the reward distribution favoured different level of competition in different contexts. This is because while in the cooperative context there is no benefit in competing, as the reward is equally shared between players, in the competitive and intermediate contexts players have a temptation to win

the trial, to avoid a loss and to boost their reward. The manipulation of the reward distribution increases the risk associated with losing by increasing the difference in reward by winner and loser (increasing $\alpha$; Supplementary Fig. 1c). It is worth noticing that in the competitive and intermediate contexts, the space dilemma is a probabilistic form of the Prisoner's Dilemma (see Supplementary Fig. 1 and methods). Each pair of participants played three block of 60 trials for each of the three contexts (cooperative, intermediate and competitive social contexts). Beyond the reward distribution shown at the start of each block of trials and the variability in players' behaviour, the three blocks of trials were visually identical, differing only in the underlying social contexts. This setup therefore allows to compare a range of cooperative and competitive behaviours across different social contexts while controlling for the sensory-motor aspects of the decision.

## Cooperativeness is shaped by the social context and the interactions within dyads

We hypothesized that participants would base their behaviour on (i) personal predispositions, (ii) the social context and (iii) the behaviour of the co-player. To demonstrate the effect of the social context, we averaged together the players' positions on different sides of the midpoint by computing the absolute distance from the closest edge, a measure of competitiveness. There was substantial variability in behaviour across all conditions, suggestive of widespread individual differences across participants (Supplementary Fig. 2). As expected, we found that the social context had a significant effect on both the average cooperativeness of players ($\beta = -0.12$, $P < 0.001$; fixed effect of condition in a linear mixed model predicting the average cooperativeness based on player and condition, see methods and Supplementary Fig 2a, b) and the absolute distance between players ($\beta = -0.25$, $P < 0.001$, fixed effect of condition in a linear mixed model predicting the average distance across players based on dyads and conditions, see methods and Supplementary Fig. 2c) suggesting that increasing the benefit of competing in a social context increased the players' competitiveness (reduced the distance from the midpoint)

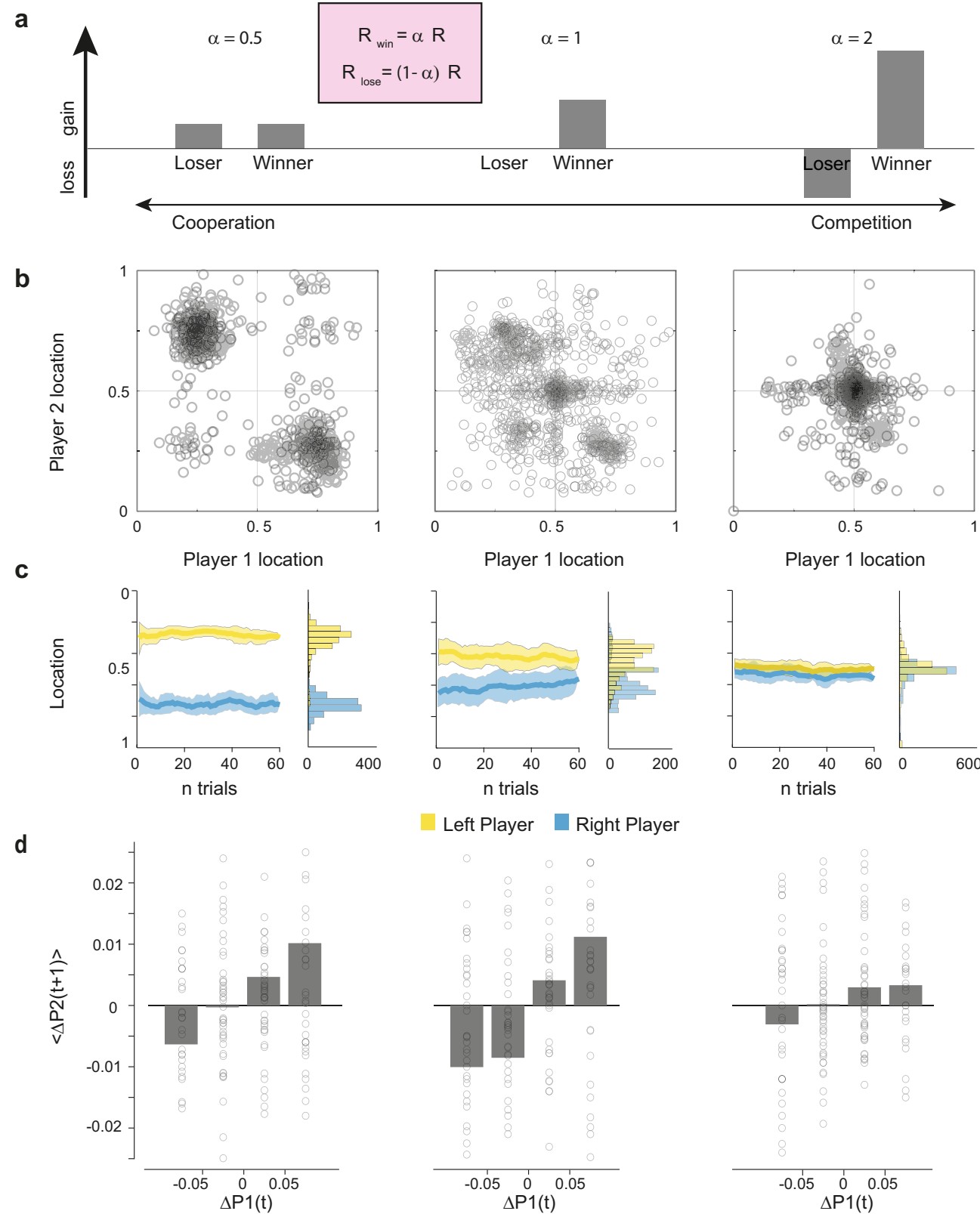

**a**
$$R_{win} = \alpha \, R$$
$$R_{lose} = (1-\alpha) \, R$$

**d** Left Player ■ Right Player

and reduced the distance among them in the space. This increase in competitiveness across contexts brought about a significant decrease in the reward collectively accumulated by the dyads ($\beta = -2.41$, $P < 0.05$, fixed effect of condition in a linear mixed model predicting the dyads reward based on dyads and contexts, see methods and Supplementary Fig. 2d) but had no significant bearing on rewards accumulated by individual participants ($\beta = -1.16$, $P = 0.26$), consistent

with the fact that competition is suboptimal for the dyad even in the competitive context while the effect on individual participants can be both positive or negative (Supplementary Fig 1 and supplementary results).

In the cooperative context, players were behaving cooperatively—positioning themselves towards the middle of one of the hemifields and sticking to one side, with a mild but significant shift towards the

**Fig. 2 | Game structure and behavioural results. a** An α parameter determines the social context and thus the amount that each player receives on each trial. The experimental design contained three social contexts that were hypothesised to shift people's competitiveness. In all contexts, the position of the closest participant to the target determined the total reward won. In the first context (cooperation), this reward would be equally shared among both players. In the second - intermediate - context, in every trial the winner takes all the reward available. In the third context, the closest player to the target wins twice the reward while the loser loses the reward from its endowment. **b, c** The strategy adopted by most participants in the cooperative context was to cooperate, and in the competitive context, to compete. In the intermediate context, participants exhibited variable responses. Responses are presented as their joint position on the *x*-axis (**b**) or over time (**c**). Kernel densities are presented on the right of each plot. Mean (bold line) and standard error (shaded area) are displayed across participants. **d** Average deviation *ΔP* (change from previous position) in a trial as a function of the co-player deviation in the previous trial. Each dot represents a participant. The co-player deviations are binned into large and small increases in cooperation/competition. In all contexts there is a tendency to reciprocate the co-player changes of behaviour in the next trial (tit-for-tat). This is particularly evident in the intermediate context, where participants were sensitive also to small increases in competition.

optimal location as time progressed ($\beta = 0.00086$, $P = 0.03$, fixed effect of trial number in a linear mixed model predicting the distance between players based on dyads and trial number, Fig. 1b, c left column, supplementary Fig. 1a, b, c). Conversely, in the competitive context, participants exhibited more competitive behaviours by positioning themselves closer to the middle of the space (Fig. 1b right) and maintaining the position during the course of the block of trials (Fig. 1c right, supplementary figure 1c). In the intermediate context, participants exhibited a range of cooperative and competitive behaviours (Fig. 1) with a significant shift from the former to the latter and convergence towards the centre as the interaction progressed ($\beta = -0.01$, $P < 0.005$, fixed effect of trial number in a linear mixed model predicting the distance between players based on dyads and trial number, see methods and Fig. 1e mid column).

Having confirmed that players' behaviours were contextually-driven, we moved on to test whether they were also driven by the co-players' behaviour. We hypothesized that participants' behavioural variability in each context could be partly explained by their co-player's behavioural variability, i.e., by deviations from their expected locations. We first looked at how players responded, on average, to changes in positions of their co-player. We grouped all changes in position from one trial to the next into 4 bins, i.e., small and large increases in cooperation or in competition. These were defined with respect to the relative position of the players: if they were converging towards the centre, this was an increase in competition. If they were instead moving away from the centre, this was an increase in cooperation (see Methods).

For all contexts, we saw that players on average reciprocated the changes of the co-player in the previous trial: if the co-player became more cooperative by moving away from the midpoint, so did the players in the next trial, whereas if the co-player became more competitive, players converged to the midpoint in the next trial (Fig. 2d). This effect was further modulated by the size of the co-player change in position: on average, larger changes of position of one player resulted in larger reciprocal changes from the co-player in the next trial ($\beta = 0.03$, $P < 0.001$, fixed effect of bin number in a linear mixed model predicting the change of position of player 2 based on bin number, condition and pairs identity, see methods and Fig. 2d). Furthermore, it was modulated by the context, being less pronounced in the competitive context ($\beta = -0.015$, $P < 0.001$, fixed effect of the interaction between bin number and condition in a linear mixed model predicting the change of position of player 2 based on bin number, condition and pairs identity, see methods and Fig. 2d). This finding suggests that players' behaviour followed a tit-for-tat strategy: they were inferring the intention of the other player, and predicting where the other player would position themselves, retaliating against co-players' increases in competition, and reciprocating co-players increases in cooperation. These effects were further modulated by the social context.

### Cooperativeness conforms to a Bayesian model

To model the behaviour in the game, including potential effects of social biases, co-player's behaviour and context on people's cooperativeness, we fitted eighteen different models (see Methods for further details). We compared different classes of models based on different

principles. The first class of models is based on the assumption that players decide their behaviour purely based on the behaviour of their counterpart, by reciprocating either their last position, their last change in position, or a combination of the two. This class of models assumes players behave in a simple reactive fashion, "titxtat" kind of behaviour, irrespective of the social context (denoted "Simple models" in Fig. 3d). A second class of models goes further in assuming that what is reciprocated is not the position of the co-player in the last trial but rather the expected position (yet unobserved) in the current trial and that the amount of reciprocation is modulated by the social context. At their cores, they all assume that a player learns to anticipate the co-player's position in a fashion that is predicted quantitatively by a Bayesian learner carrying out the same task ("Bayesian models" in Fig. 3a–d). They also assume that this expectation is reciprocated in a titxtat fashion. However they differ in how this expectation is mapped onto a choice, allowing for different degrees of influence of the context, their counterpart's behaviour and the player's own bias. A third class of models assumes that participants were choosing what to do based not only on the other player behaviour but also on the outcome of each trial, with different assumptions on how winning a trial should change their behaviour in the next (becoming more or less cooperative). This class of models is effectively assuming that the player behaviour would be shaped by the reward collected ("Reward models" in Fig. 3d). We used formal Bayesian model comparison (see Methods) to identify the best-fitting model (Fig. 3d). The winning model is a Bayesian model and contained features that accounted for both people's biases towards cooperativeness, how the behaviour of the other player influenced subsequent choices and the influence of the social context.

All Bayesian models significantly outperformed both the simple reactive models and the reward-based ones. To validate this modelling approach and confirm that players were trying to predict others' positions rather than just reciprocating preceding choices, we ran a regressions model to explain participants' choices based on both the last position of the co-player and its Bayesian expectation in the following trial. We found that expected positions were significantly better predictors than preceding choices (see Supplementary Fig. 6b). Both these pieces of evidence point to the fact that whilst players implement tit-x-tat strategies, they do so in a way that considers all past behaviour of their co-player, effectively discounting their latest choice with prior decisions, therefore being more robust to single, potentially accidental, deviations if there was a consistent history of cooperation.

Specifically, the winning model (B6) implemented (i) a "tit-for-tat" strategy whereby the first player reciprocated the co-player's expected choice in their own hemifield by cooperating by the same amount scaled by a "TitxTat" factor (Fig. 3a); (ii), this factor was determined by a parameter normalized by a context-dependent factor inversely proportional to the increase in social risk associated with the redistribution parameter α, i.e., the higher the redistribution, the lower the risk associated with losing, the higher the TitXTat factor (Fig. 3a, f); (iii) a social bias parameter determining individual inherent preferences towards competing or cooperating ("Social Bias"; Fig. 3a, f); (iv) a parameter capturing players' Precision (e.g., players may press the button too early or too late compared to the location they aim for), increasing their

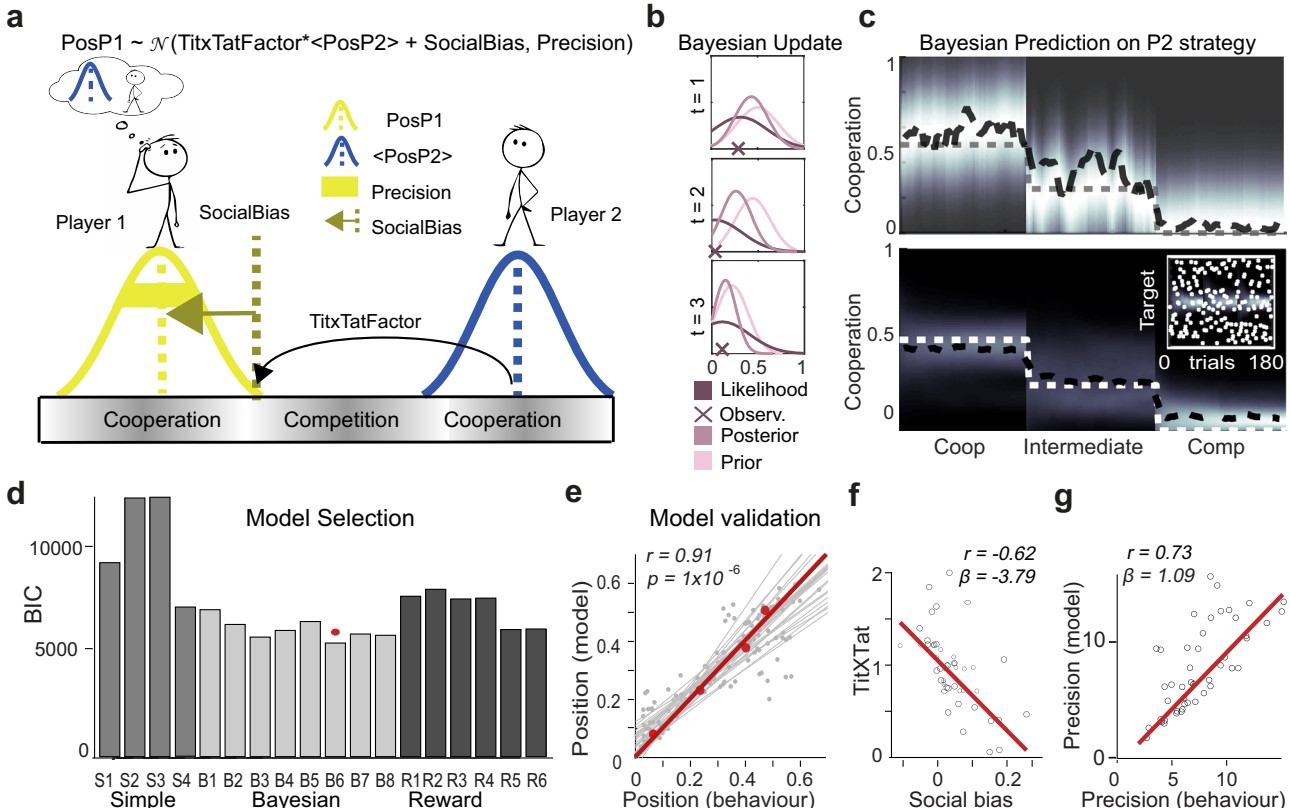

**Fig. 3 | Model predictions. a** Our best model described how player 1's (P1) choices resulted from (i) a tit-for-tat strategy whereby P1 reciprocated player 2's (P2) anticipated position ("Exp_PosP2") scaled by a context-dependant tit-for-tat factor (ii) P1's social preference or bias ("SocialBias") and (iii) a precision parameter capturing players' ability to choose the desired location. **b** Representative examples of 3 trials during which P1 learned to anticipate P2's position following Bayesian learning: observing P2's position on a given trial updates P1's belief about P2's strategy. **c** Top figure: Single participant representative example of P1's positions explained by the anticipated position of P2 following the Bayesian learning procedure described in **b**. Coop: cooperation; comp: competition; intermediate context as described in fig1. **Bottom figure:** Population averages. In the inset, anticipated position of the target vs actual position: as expected, a Bayesian learner cannot predict the position of the random target. Bayesian models B4-B5 included a prediction about the target location. **d** Bar plots illustrating the results of the summed integrated Bayesian Model Selection. Lower BIC scores indicate better fit.

Models are divided into three classes, 'Simple' (S1–4), 'Bayesian' (B1–8) and 'Reward' (R1–6) based on their underlying logic (see text). The Bayesian model B6 with context-modulated tit-for-tat and "SocialBias" performs best (BIC = 5553). **e** Scatterplot showing linear correlation between empirical and predicted choice positions (r: Pearson's correlation coefficient, $N = 4$ bins x 50 participants = 200. One-sided $P$ of correlation as large as $r$). For each participant, positions were binned in four bins and the average model prediction for each bin was computed. Grey dots are individual participant bin averages. Red dots are population averages. Grey lines reflect individual participant fits, the red line is the fit of the population averages. **f** Scatterplot showing linear correlation between the "tit-for-tat" and social bias parameter. Each dot is a participant ($N = 50$). **g** Scatterplot showing the linear correlation between the precision parameter and the individual behavioural precision estimated by the inverse of the standard deviation of P1's positions observed during the game. Note that all participants served as P1 in the analyses ($N = 50$).

variability in behaviour beyond the one that can be explained by the social context and the co-player behaviour. Thus their actual choice is normally distributed around the "titXtat + SocialBias position" (with the standard deviations being a model parameter). Two other Bayesian models (B7-B8) fitted the data slightly better than model B6. These models used an additional parameter to estimate the probability that a co-player might "betray" by arbitrarily becoming more competitive. This probability is estimated in a Bayesian fashion based on the history of unexpected deviations. However, the inclusion of the extra parameter (which increases the BIC) is not justified by a small improvement in negative log likelihood suggesting that it is unlikely that our players encoded the probability of betrayal independently of the effect of context (which makes participant more cautious−less cooperative−anyway). In any case these models are inherently similar and make very similar behavioural predictions, since they share the same Bayesian architecture and three free parameters.

We found that observed and predicted positions from the winning model were significantly correlated (Fig. 3e, Pearson's correlation coefficient $r = 0.91$, $P = 1 \times 10^{-6}$, see Methods and Supplementary Fig. 3a, b for individual and averaged participants fit). Moreover, note that

all parameters of the winning model fitted to behaviours revealed significant variability (Supplementary Fig. 4c). This is important because we can then explain the variability in our participants' responses with the variability captured by our model parameters. As such, using a regression analysis, we found that the precision fit by the model (see Methods) was significantly correlated with the variance of player 1's positions observed during the game ($\beta = 6.64$, $P = 1 \times 10^{-6}$, Fig. 3g).

We subsequently examined the relationship between the other parameters. If participants vary in how much they adapt to the other players, but also vary in their initial bias along the cooperation-competition spectrum, we would predict a relationship between participants' parameters in the model. Strikingly, we found a strong negative correlation between participant's TitXtat parameters and their social biases ($r = −0.62$; $\beta = −3.79$; $P < 0.05$), suggesting that participants distributed along an axis, with, on one end, participants who were more inclined to be cooperative irrespective of what the other player was doing, and, on the other, participants whose behaviour was more flexible and dependant on the co-players' behaviour. Importantly this anticorrelation was not derived from the specific model we used.

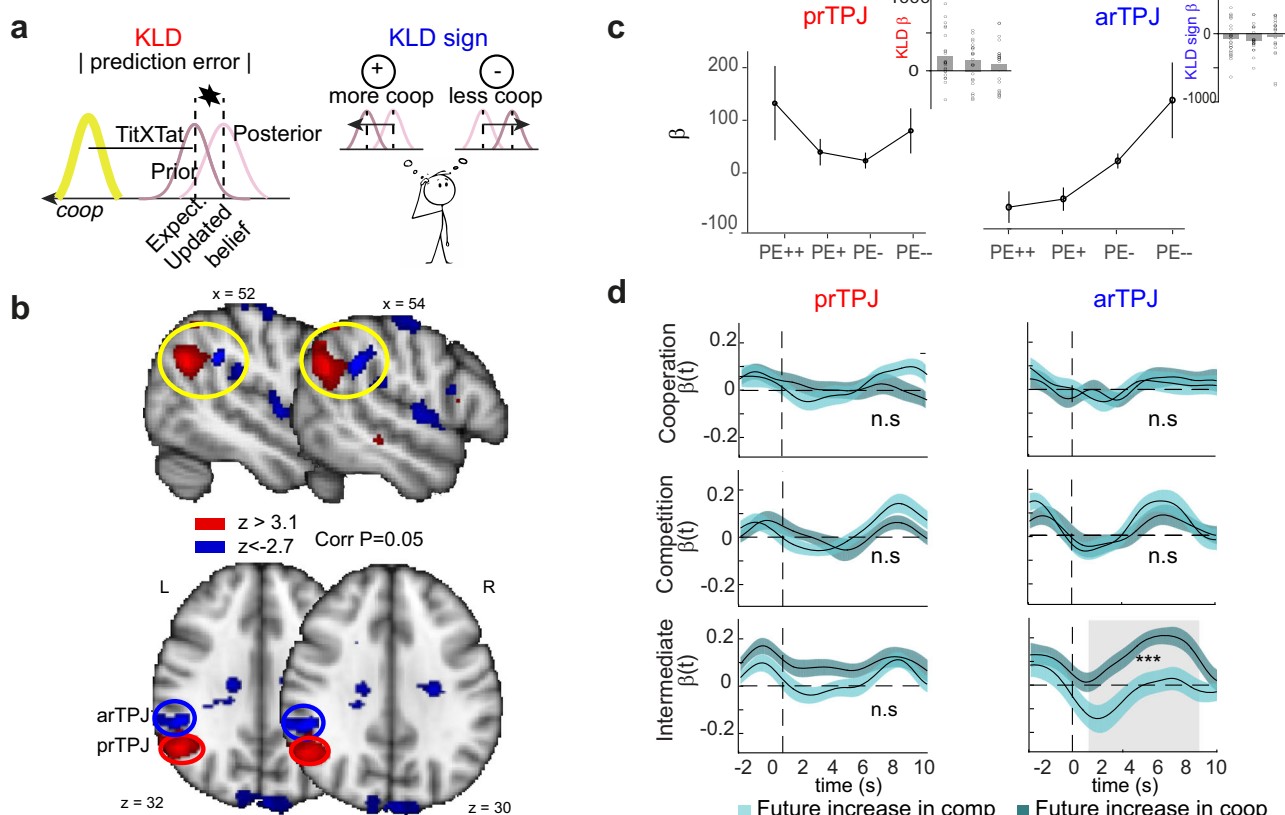

**Fig. 4 | The TPJ encodes all dimensions of a social prediction error and predicts future behaviour. a Left panel:** The magnitude of the KLD represents the absolute prediction error and this is to what extent P2 position deviated from what was expected. This is a learning signal informing how much the P1 needs to adjust his position on the next trial. **Right panel:** The sign of the KLD represents the direction of the violation of expectation: was P2 more cooperative than expected or more competitive? **b** One whole-brain analysis parametrically tested for voxels where activity correlated with the trial-by-trial estimates of (i) the KLD magnitude (in red) and (ii) the KLD sign (in blue). **c** Average (+/− SEM) population βs for GLM 3 in prTPJ (left) and arTPJ (right) across four groups of trials binned based on their value of KLD magnitude and sign (ordered from positive prediction errors signalling big increases in cooperation−PE++ - to negative prediction errors signalling big increases in competition−PE−). In the insets, the βs from GLM 1 show that prTPJ and arTPJ are encoding KLD and KLD sign across all contexts. **d** Times series analyses revealed that, depending on whether, in the next trial, the participant became more cooperative or competitive (measured through the sign of the change in position from the previous trial), the activity in arTPJ would be different, with a higher signal when participants became more cooperative in the intermediate condition. Traces are population averages (+/− SEM).

This anticorrelation between social bias and TitXTat was also found when fitting a simple linear model which was predicting players' positions based on the co-players' position and a constant term: the linear term was anticorrelated with the constant term for all conditions ($\beta = -0.62$, $P < 0.001$ for condition 1; $\beta = -0.95$, $P < 0.001$ for condition 2; $\beta = -1.88$, $P < 0.001$ for condition 3), suggesting that the trade-off between pro-sociality and tit-for-tat like behaviour is a feature of participants' behaviour, that can be accounted for by the model.

**Different portions of TPJ encode a social prediction error**

We hypothesised that people will change behaviour based on the social context, will show a range of social biases and will update their behaviour based on their interactions with the other player. Having demonstrated that a Bayesian model can capture such behavioural effects in the Space Dilemma, we next examined if neural signals might similarly reflect the model. Within the model, a key component is tracking the behaviour of the other person, that is, predicting how competitive someone is and then observing the other person's behaviour. In a Bayesian framework such tracking occurs through Kullback-Liebler divergence (KLD), which quantifies a social prediction error− the difference between the expected location of the other player and their actual location (Fig. 4a, top panel). Given previous evidence that unsigned prediction errors (the absolute magnitude of the error or "surprise" regardless of direction) and signed prediction errors

(positive when something is higher than expected and negative when something is lower than expected) may be dissociable[51–54], we included in our main GLM (see methods) two parametric regressors coding the unsigned (magnitude of difference between expected location of P2 and actual location) and signed KLD (positive magnitude when P2s location is more cooperative than expected and negative when P2s location is more competitive than expected, Fig. 4a bottom panel) and examined responses time-locked to when the other player response was revealed to the participant in the scanner (Fig. 4a top panel in blue).

A whole-brain analysis revealed significant activity in the right TPJ reflecting both components but in distinct sub-regions. The unsigned prediction error was represented in a posterior portion of the right TPJ (Fig. 4b, prTPJ $Z = 4.40$, MNI: $x = 52$, $y = -58$, $z = 30$) while the signed prediction error was encoded in a contiguous cluster in the anterior part of the right TPJ (Fig. 4b, arTPJ; peak $Z = -3.67$, MNI: $x = 50$, $y = -38$, $z = 32$). Both regions survived multiple comparison correction (Fig. 4b; $Z > 3.1$ cluster forming threshold, whole-brain cluster-based correction $P < 0.05$; GLM1). To test the full parametric effect of the two clusters in TPJ we run a control GLM (see methods) to test how their activity varies across four different groups of trial split based on the KLD value and its sign. This ROI analysis reveald that arTPJ activity increased with value of prediction errors signalling increases in competition of the co-player (Fig. 4c, right) whilst activations in prTPJ show a *u*-shaped

relationship (Fig. 4c, left), providing additional independent evidence that these two sub-clusters in TPJ encode the sign and the absolute value of the prediction error, respectively. Additionally responses to the magnitude of the Social Prediction Error were also found in the Inferior Frontal Gyrus (IFG; 50, 16, 14), Middle Frontal Gyrus (MFG; 44, 16, 40), bilateral Insula (INS; ±34, 22, −4) and in the Middle Temporal Gyrus (56, −30, −10/60, 4, −24). In all contexts (see insets, Fig. 4c), these regions appear to encode signals linked to the surprise occurring when observing the co-player's location (and thus the degree of competitiveness experienced) and contrasting it to the expectation based on their past behaviour.

The results above suggest that two regions of the TPJ may encode the surprise and signed prediction error associated with the other player's competitiveness in all contexts, but does this relate to how participants changed their behaviour? To address this question, we looked at how trial by trial changes in the amplitude of the neural signals were linked to behavioural changes in the following trial. We extracted the time-courses of the BOLD (Blood-Oxygen-Level-Dependent) signal in these two TPJ regions at the time of the other player's response and examined whether signals on a trial were predictive of a change in behaviour (an increase or decrease in competitiveness) on trial ($t + 1$) (see Methods). We found a correlation between change in behaviour and signals in the arTPJ ($P < 0.001$, $t$-test fMRI betas at corresponding time points Fig. 4d) in the intermediate context. This condition is the one in which there is the greatest variability from trial to trial in behaviour and thus where monitoring the other player's responses to guide one's own is the most important. Thus, whilst the prTPJ signals how surprised one is about another's competitiveness, the arTPJ encodes a directionally specific prediction error that is predictive of a future change in cooperativeness in the context where it is most important for people to understand when to do so.

## Social context modulates how posterior dorsomedial frontal, cingulate and paracingulate cortices encode updates to cooperativeness for self and other

Whilst the TPJ encoded surprise signals across all social contexts, how does the social context modulate neural activity in order to update behaviour? To test this we compared the neural activity across social contexts to identify whether any regions were engaged differently when changing cooperativeness or monitoring another's changes in cooperativeness. We looked at the contrast between the two extreme social contexts: the cooperative and competitive ones. In our main whole-brain analysis (GLM I, see methods) we parametrically tested which parts of the brain encode (i) P1's changes in the level of cooperation at time of decision (*self coop*) and (ii) the sign and magnitude of the social prediction error (P1's surprise about P2 changes of position− *another coop*) at the time when P2 position is revealed. We then ran a contrast analysis between the cooperation and competition contexts. All the reported activation clusters were identified with an uncorrected threshold of $P < 0.001$ and corrected for family-wise error (FWE) at the cluster level at $P < 0.05$. GLM II; Fig. 5.

We found a significant difference in activity in a cluster in the posterior portion of the dorsomedial prefrontal cortex (pDMPFC) extending posteriorly across the pre-SMA and inferiorly to the cingulate cortex (pDMPFC, $Z = −4.09$, MNI: $x = −8$, $y = 16$, $z = 52$), where individual changes in cooperation levels where encoded differently between the cooperative and competitive contexts at the time when the participant was choosing how cooperative to be (*self coop*; Fig. 5a). Additionally, we found significant difference between cooperation and competition in the activation of an area in the Superior Frontal Gyrus (SFG; MNI: $Z = −3.54$, 28, 6, 56), in the right Insula (INS; MNI: $Z = −3.85$, 30, 26, 0) and in the Precuneus (PC; MNI: $Z = −3.95$, −6, −56, 56).

Furthermore, we found two regions which showed a significantly different activation between cooperation and competition for the sign of the social prediction error. The first was lying in the anterior cingulate gyrus (ACCg, $Z = −3.13$, MNI: $x = 0$, $y = 34$, $z = 20$, Fig. 5b) while the second was approximately located in the paracingulate gyrus in Broadmann area 32 extending to the cingulate and anterior dorso-medial prefrontal cortex (PaCg, $Z = −3.36$, MNI: $x = 2$, $y = 50$, $z = 12$)− both were active at the time in which the opponent response was revealed, showing a significant difference in the way it encoded the sign of the social prediction error between the competitive and cooperative contexts.

Interestingly, these last two areas positively signalled increases in cooperativeness of the co-player during the competitive context, but it negatively signalled increases of cooperativeness during the cooperative context (Fig. 5c). The results above suggest that these areas not only encode the sign of social prediction errors, but they do it differently in different social contexts. If the social context modulates their activity, it is possible that cingulate and paracingulate cortices, contrary to the arTPJ, might have different roles in different contexts. How does this relate to behaviour?

To address this question, we looked at how trial by trial changes in the amplitude of the neural signals in these two clusters were linked to behavioural changes in the following trial. Once again, we extracted the time-courses of the BOLD signal in these two regions at the time of the other player's response and examined whether signals during a given trial ($t$) were predictive of a change in behaviour (an increase or decrease in competitiveness, i.e., change in location) on the subsequent trial ($t + 1$) (see Methods). For both region, we found a correlation between their activity and change in behaviour ($P < 0.001$, $t$-test fMRI betas at corresponding time points Fig. 4d) in the intermediate and in the competitive condition. Interestingly, and consistently with the idea that the social context modulates their role, both areas appear to predict increases in cooperation in the intermediate condition and increases in competition in the competitive condition. Thus, both clusters are predictive of a future change in competitiveness but in different ways in different contexts.

Finally, we reasoned that if these areas are significantly involved in determining behaviour, this should be reflected by the parameters of our model. We therefore correlated the parameter representing the *social bias*, capturing the degree to which participants' behaviour was biased towards cooperation with the average *betas* of the two clusters for the sign of the social prediction error at the time the co-player response is revealed. We performed the same analysis for the player increases in cooperation at time of response. Similarly, we did the same for the *titXtat* parameter, capturing the degree to which participants' behaviour was determined by the attempt to reciprocate the level of cooperation of the co-player.

Intriguingly, we found that the representation of increases of cooperation for self positively correlated with the *social bias* parameter and anticorrelated with the titXtat parameter for both clusters (Supplementary Fig. 5e). Furthermore, a detailed analysis of how the representation of self and other cooperation changes across contiguous clusters in ACC backed up by correlation with model parameters lend some evidence to the existence of a self-other gradient along the rostro-caudal axis within ACC (Supplementary results and supplementary Fig. 5d–f). Taken together, these results provide strong evidence that the cingulate and paracingulate cortex are instrumental in adjusting behaviour in response to the action of partners/competitors and according to the social context.

## Discussion

Competition and cooperation are two social orientations that can either hamper or facilitate individual achievements. While traditionally cooperation and competition have been studied separately, they are not all or nothing but occur along a continuum. Using a new economic game, modelling and fMRI we revealed some of the computational and neural mechanisms controlling the trade-off between competition and cooperation. Using a continuous spatial location as a parametric

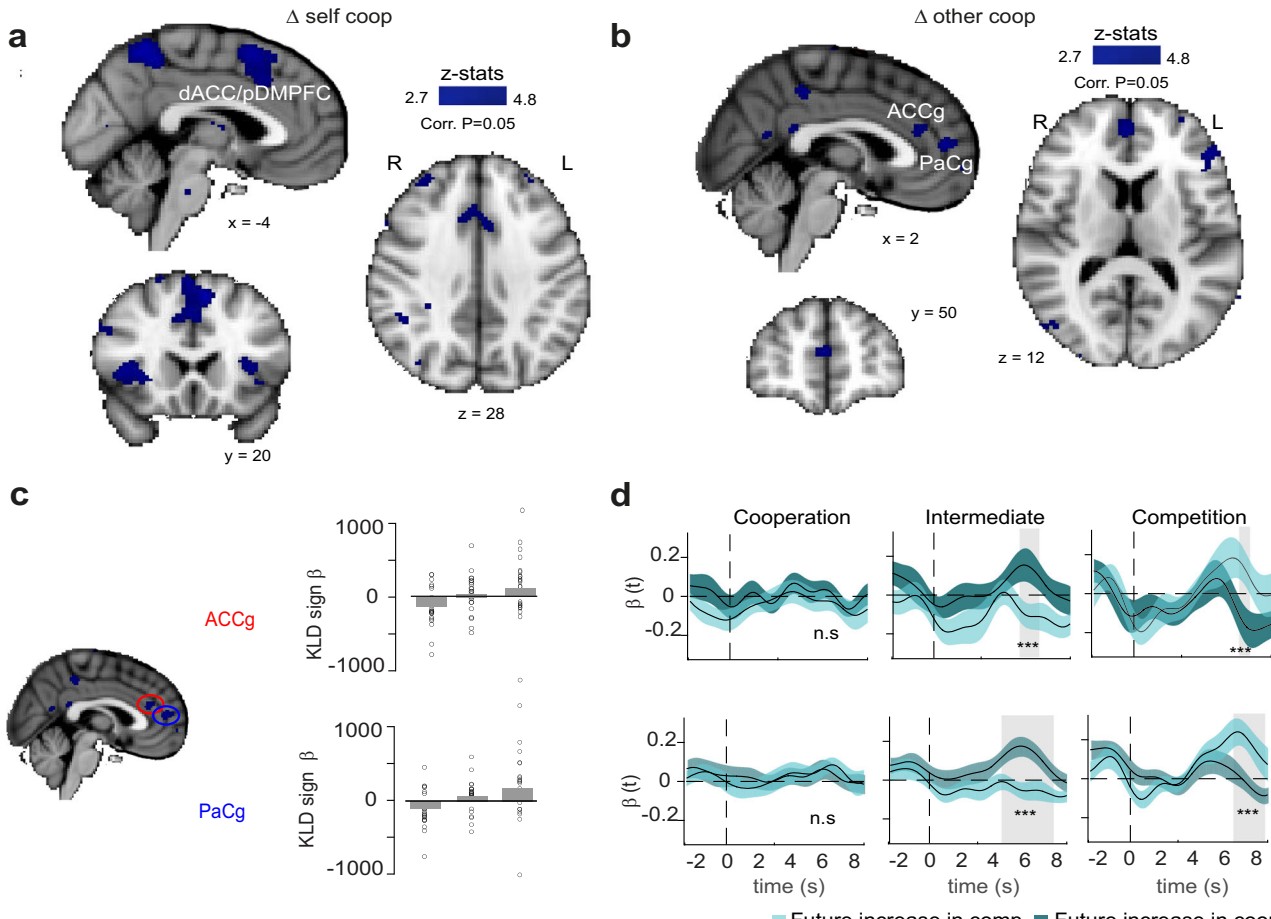

**Fig. 5 | Social context modulated medial frontal, cingulate and paracingulate cortices encoding of changes in self and others cooperativeness. a** This whole-brain analysis parametrically tested for voxels where the social context modulated how activity correlated with the trial-by-trial estimates of P1's changes in the level of cooperation at the time P1's response. **b** This whole-brain analysis parametrically tested for voxels where the social context modulated how activity correlated with the trial-by-trial estimates of the sign of the social prediction error (P1's surprise about P2 changes of position) at the time when P2 position is revealed. **c** βs showing the strength of ACCg and PaCg encoding of KLD sign across social contexts (from cooperative to competitive; bars represent population averages, $N = 25$). **d** Time series analyses revealed that, depending on whether, in the next trial, the participant became more cooperative or competitive, the activity would be different, with a higher signal when participants became more cooperative in the intermediate condition. Traces are population averages (+/− SEM).

measure of the cooperation-competition continuum, we showed that our new paradigm allows to explore a range of cooperative and competitive behaviours and to compare them across different social contexts while controlling for the sensory-motor aspects of the decision.

We showed that people's degree of cooperativeness is shaped by (i) what the social context favours, (ii) the nature of the interaction between two individuals and (iii) predispositions towards cooperativeness regardless of the context or the other player's behaviour. These patterns of behaviour were captured by a Bayesian model, which included parameters weighting the social context, the participants' social bias, and dictating how much the other player's actions were influencing one's own. Our results point to the important role of the rTPJ in coding social prediction errors that lead to subsequent changes in competitiveness. We also found that distinct regions of the medial prefrontal, cingulate and paracingulate cortices coded information linked to how the social context and one's social bias shape people's own cooperativeness as well as the monitoring of other people's one.

Understanding which social contexts facilitate cooperative behaviours is of paramount importance for human societies, both to increase well-being and reduce conflicts. Conversely, understanding how to control and constrain competitive behaviours can be beneficial to improve the performance of a group and its benefit to the wider society. Much research has investigated cooperative behaviour in

games like the Prisoner's Dilemma, the Chicken's Game and the Trust Game[9,11,31]. Conversely, competition has been studied in zero-sum games like the matching pennies[55–58]. While a few experimental and theoretical studies have examined non-binary versions of Prisoner's Dilemmas[22,59,60] and explored the impact of changing the cost/benefits of cooperation[61], very few studies have attempted to directly compare these two social orientations and those that have done so[62,63], did not consider that their trade-off occurs along a continuum. To our knowledge, this study is the first one to propose a continuous parametrization of the competition-cooperation axis to study the computational and neural mechanisms underlying the continuous trade-off between the two strategies.

Studying the continuous nature of the cooperation-competition trade-off is important for several reasons. First, our paradigm affords a richer and more flexible behavioural repertoire of social approaches, as it allows observing fine-tuned changes in behaviour that would otherwise remain "latent" in a binary setting. This is particularly important in the context of social interactions, as minor adjustments of behaviour are observable and can lead to shifts in strategy, inducing social dynamics that could remain undetected in a binary setting. For instance, in our intermediate condition we see the players slowly drifting towards the more competitive position, due to a cycle of fine adjustments reflective of a combination of titxtat and the rational

incentive to win the game. Second, a continuous set up is important to identify the neural activity underpinning the behaviour. In binary choice tasks, both strong and weak intentions to cooperate might be reflected in identical choices being made before a sudden shift in policy occur. This would bear the inference of a large prediction error, obfuscating the latent drift towards a new social orientation. Thus the gradation of prediction error values that best accounted for shifts in people's behaviour and intentions to compete would not be well predicted by a single observed behaviour.

Decades of research have linked the TPJ to the processing of social information, both in cooperative and competitive contexts[63–65]. Classical accounts have posited that this region had a role in distinguishing one's own mental states from others, as revealed by the BOLD response changing when another's beliefs are revealed to be erroneous and different from one's own[40]. More recent theoretical work has argued for a social predictive coding framework for theory of mind postulating errors signals in TPJ[66] and previous studies have reported evidence of rTPJ encoding of both unsigned[67–70] and signed prediction errors[71] in a range of social contexts and behaviours. rTPJ has also been implicated in tracking the expectation of cooperation of others in public good games[72,73]. Our results show that this region does indeed signal errors during cooperative interactions but go beyond classical accounts in several ways.

Specifically, we show that when monitoring another's behaviour its activity covaried with the Bayesian update of the expectation about the co-player's intention to cooperate, thus quantifying the prediction error in a social interaction. This prediction error was signalling the discrepancy between how cooperative someone else had behaved, compared to an expectation of the degree of cooperativeness they would exhibit. In addition, we distinguished between signed and unsigned prediction errors localized to distinct sub-regions within the TPJ, the prTPJ and arTPJ respectively. Lastly, we showed that the signed prediction error signalling in the arTPJ also correlated with subsequent changes in how cooperative a person would be in the next trials. All of these findings converge on the notion that the TPJ plays an important role when flexibly adapting one's behaviour during social cooperative interactions, an important component of mentalizing.

However, our findings dissociate the contributions of TPJ sub-regions, with the prTPJ signalling how surprised one is at another's competitiveness and the arTPJ further translating this into a directionally specific code that is used to shift one's own cooperativeness in the future. This organisation affords the flexibility for the prediction error to be attributed directly to the process of extracting others' intentions and be used to help the player select the optimal response in subsequent trials. Such an interpretation accords with anatomical evidence that the TPJ contains distinct sub-regions that have distinct functional roles. The two clusters we identified in the TPJ overlap with two distinct regions as identified with resting-state MRI and diffusion-weighted parcellations[74]. Although there has been some suggestion that the prTPJ is the sub-region most strongly associated with social cognition[75], by using a more refined, continuous task, and a Bayesian model, we show that both sub-regions may compute important information for social cognition. Previous studies have shown the prTPJ to signal prediction errors during iterative economic games[37] and when evaluating how trustworthy another's advice is[32]. However, such studies did not have a task where competitiveness occurred along a continuum, nor distinguish between signed and unsigned prediction errors.

Work examining computations outside of social cognition has highlighted the importance of distinguishing between signed and unsigned prediction errors[54,76,77]. Unsigned prediction errors are crucial for signalling the salience and thus importance for attending to information, but do not carry valence information that is useful for adapting behaviour[52,53,78]. In contrast, signed prediction errors may be important for subsequently updating behaviour, up-regulating or inhibiting behaviours that did or did not lead to a desired outcome. Such signals have previously been dissociated in non-social tasks, with signed prediction errors in medial frontal cortex putatively important in updating models and expectations of future events[79]. However, this distinction has rarely been made in social cognition research. Here, we show dissociable signed and unsigned prediction errors in discrete TPJ zones that enable someone to attend and flexibly update behaviour across different social contexts, based on how much more cooperative or competitive a person was than expected.

In addition to the TPJ, we also found sub-regions of the medial frontal, cingulate and paracingulate cortex previously linked to social cognition that encoded several other features of our model and behaviour[29,37,38,42]. The findings indicate roles across the medial frontal cortex for carrying information about one's social biases and adaptability to others' competitiveness, as well as shifting responses depending on the social context. In particular, we found a cluster in the posterior DMPFC extending inferiorly to the cingulate cortex where individual changes in cooperation levels where encoded differently between the cooperative and competitive contexts. We also found a region in the anterior dmPFC—putatively in paracingulate cortex (PaCg)—that signalled a social prediction error when monitoring another's player, which was also linked to the updating of one's behaviour along the cooperation-competition continuum as well as correlating with variability in the social bias towards cooperation or competition across contexts.

Such a role aligns with work implicating this region in the processing of social influence[80] as well as classical research implicating this region in mentalizing processes. Previous work has shown this region signals prediction errors when shifting one's preferences to align with other peoples[36,80–82], contains individual neurons which signal when others' behaviour is erroneous[47]. Moreover, individual differences in activity of this region have also been linked to the degree to which one conforms to social norms in economic decisions[83]. In addition, a plethora of classic research shows that this region is involved in processing and inferring others mental states during false-belief tasks, and computational tasks where one processes levels of trust in others and when processing others' actions during economic games[84–86].

Our results therefore support an emerging view of the paracingulate cortex as playing important roles in processing others' intentions, with variation in its response linked to variability in the extent to which people choose to shift their behaviour in responses to others. However, here we show that responses in this region may be involved in inferring others' intentions, and updating those predictions through prediction errors, when deciding how much more cooperative or competitive to be. Moreover, they suggest that this region multiplexes several different pieces of information that influence one's position along an axis, including one's bias towards cooperation, the influence of the social context and inferences about the intentions of others.

In addition to the paracingulate, we also found context-dependant signals encoding social prediction errors in a neighbouring region lying in the ACCg. Specifically, we found that, similarly to PaCG, ACCg activity correlated with the sign of the prediction error in a way which was modulated by the social context, signalling increases of cooperation in the competitive context and increases of competition in the cooperative one. ACCg activity was also linked to the updating of one's behaviour in the next trial as well as correlating with individual variability in the social bias parameter of our model, capturing the intrinsic propensity of being more or less cooperative. There is growing evidence that this region is engaged when processing specifically social information[32,38,44,87], and particularly in signalling predictions when expectations about others are violated[43,45,46,88] or tracking the likelihood of defecting cooperation in a public goods game[72]. Our results support the notion that the ACCg carries social prediction error

signals, however, existing work had typically shown that these signals were important for learning through observation, for correcting others mistakes, for identifying others' erroneous predictions or for identifying whether to trust another. Here we show that such signals also demonstrate how unexpected someone else's degree of competitiveness is, and that such signals change depending on whether the social context is one that favours cooperation or competition. Thus, our results suggest that social prediction errors in the ACCg may be used to understand how motivated another person is to obtain benefits for themselves[38], but does so in manner that differs depending on the social environment and that correlates with future changes in behaviour.

Over the last decade there has been a shift in focus towards identifying the computational mechanisms that guide social behaviour[89]. Much of this work has begun to show that models based around reinforcement learning and Bayesian principles may provide the framework that scaffolds social information processing. Previous work has identified social prediction errors that underlie prosocial behaviour[90], teaching[43], trust[32], mentalizing[37], false belief processing[45,91] and a range of other processing requiring social learning[12,35,92]. Our results concord with the notion that Bayesian principles and prediction errors can guide social behaviour and socio-cognitive processes.

In this work, we developed an economic game that generalizes the Prisoner's Dilemma[49,50] into a continuous measure and reproduce a well-known economic principle of locational equilibrium in duopoly described in the Hotelling law[48]. On a broader level this work converges two lines of research exploring social behaviour, work in behavioural economics using economic games[20,93,94], with those arguing that human decision-making may be best understood with approaches from foraging theory[95,96]. In fact, the task could also be framed as a foraging problem where one had to position oneself in a location. The co-player was likely to be treated as a potential 'predator' in the competitive context when the reward of a player correspond to a loss for the co-player, but not in the cooperative one, with such behavioural flexibility linked to mPFC responses. Such findings relate to research which has suggested that activity in several mPFC sub-regions may be encoding the proximity of threats in the environment. Our results somewhat concord with this notion, but suggest that rather than proximity to threat, several mPFC regions are involved in integrating one's overall preference for cooperativeness, the changes in social context and information about the actions of the other player. All of these signals are necessary to identify where a player will position themselves, as well as being processes responsible for judging where the other player will position themselves. As such, these regions may be engaged when potentially close to threats by processing information that allows one to adapt behaviours accordingly.

Ultimately, our paradigm allows exploring how the social context changes the involvement of the neural network involved in arbitrating the competition-cooperation trade-off. Future experiments using this paradigm might help to probe further hypotheses. For example, future studies might address questions relating to the impact of uncertainty, for instance varying the reward probability of each location (making the reward location– to an extent – predictable through uni/multi-modal distribution instead of a uniform one), the difficulty of the task (speeding up or making it harder to make a certain choice) or the social dynamics (increasing the number of players or the distribution of the rewards among them).

In conclusion, we used a new economic game–the Space Dilemma –that allows people to be cooperative or competitive along a continuum. We show that people's level of cooperation is dependent on several sources of information, including the behaviours favoured by the structure of the environment, their own biases towards cooperation, and online updating based on the competitiveness of another player. We show that such behaviour can be approximated by a Bayesian learner, including parameters that scaled each of these features impacting behaviour, with signals in the TPJ, mPFC, ACCg and PaCg−regions previously implicated in social cognition−processing the information that guided behaviour and signalling social prediction errors when monitoring the other player's competeness. These findings shed light on the multiple features that guide how cooperative we want to be, and how we shift our behaviour along a continuum.

## Methods

### Participants

The study complied with all relevant ethical regulations. The study protocol was approved by the Institute of Neuroscience and Psychology Ethics Committee at the University of Glasgow. Written informed consent was obtained in accordance with the Institute of Neuroscience and Psychology Ethics Committee at the University of Glasgow. Twenty-seven same-sex pairs of adult human participants participated in the fMRI experiment. This number was determined based on a priori estimates of sample size necessary to ensure replicability on a task of similar length[97]. All were recruited from the participants' database of the department of Psychology at the University of Glasgow. For each couple one participant was in the scanner and the other in an adjacent room. Two pairs were removed from the analysis: one for excessive head movements inside the scanner, the other for a technical problem with the scanner. The remaining couple of participants (7 of males, 18 of females), were all right handed, had normal or corrected-to-normal vision and reported no history of psychiatric, neurological or major medical problems, and were free of psychoactive medications at the time of the study.

### Stimuli and behavioural task

All participants played the Space Dilemma in pairs of two. Before starting the game they were given a set of instructions explaining that they had to imagine that they were foraging for food in a territory and asked to make a prediction about the position of the food (a straight line that represents the territory, Fig. 1). They were told that in each trial the target "food" would appear somewhere in the territory as its position is randomly sampled from a predefined uniform probability distribution. They were shown examples of possible outcomes of a trial (Fig. 1) and they were given information about the conditions of the game. During the game, in each trial, they were presented with a bar moving across the space (representing their location) and asked to commit to a location by pressing a button while the bar passes through it while moving in the linear space. Participants therefore choose their locations in the space through the timing of a button press. They indicated their choice by pressing one of three buttons on a response box. The bar takes 4 s to move from one end to the other end of the space. Once stopped, it remains at the chosen location for the remainder of the 4 s. This location signalled their prediction about the target position. The two participants played simultaneously, making first their predictions and then watching the other player's responses (for 1–1.5 s). After both players had responded, the target would be shown (for 1.5 s). Inter-trial intervals were 2–2.5 s long. At any trial, the participant who made the best prediction (minimising the distance $d$ to the target) was indicated as the trial's winner through the colour of the target, obtaining a reward which would depend on the distance to the target: the shorter the distance the higher the reward. In the rare circumstance where players were equidistant from the target such reward was split in half between the two players who were both winners in the trial.

In order to enforce different social contexts we introduced a reward distribution rule whereby each trial reward would be shared between the winner and the loser according to the rule

$$R = (1 - \min(d)) \qquad (1)$$

$$R_{win} = \alpha R; \; R_{lose} = (1-\alpha)R \qquad (2)$$

Where $\alpha$ is a trade-off factor controlling the redistribution between winners and losers in each trial. By redistributing the reward between winner and loser the latter would also benefit from the co-player minimising their distance to the target. Increasing the amount of redistribution (decreasing $\alpha$ below 1) constitutes an incentive to work out a cooperative strategy to decrease the average distance of the winner from the target (that is, irrespective of who the winner is) and therefore increase the reward available in each trial which would be redistributed. Decreasing the amount of redistribution can instead lead to punishment for the losers (increasing alpha above 1) adding an incentive to compete to win the trial.

All participants first participated in a behavioural session where they were randomly coupled with one another and played three sessions of the game in three different conditions specified by the value of the trade-off factor $\alpha$. In the first condition ($\alpha = 0.5$, cooperative condition), the reward was shared equally between the two players, irrespective of the winner. In the second condition, the winner gets twice the amount of the reward ($\alpha = 2$, competitive condition), while the other player will lose from their initial stock an amount equivalent to the reward. In the third condition, the winner will get the full amount of the reward and the other will get nothing ($\alpha = 1$, *intermediate condition*). The participants were instructed about the different reward distribution (through a panel similar to Fig. 2c). In total, participants played 60 trials in each of the three conditions for a total of 180 trials.

At the end of the behavioural session, participants were then asked to fill in a questionnaire where their understanding of the game was assessed together with their social value orientation[98]. If they showed to have understood the task and were eligible for fMRI scanning they were later invited to the fMRI session which occurred 1–3 weeks later. In total, 81 participants took part in the behavioural session and 54 participated to the fMRI session.

In the fMRI sessions, participants were matched with an unfamiliar co-player they had not played with in the behavioural session and it was emphasised not to assume anything about their behaviour in the game. We did not use deception: participants briefly met before the experiment when a coin toss determined who would go into the scanner and who would play the game in a room adjacent to the fMRI control room. Both in the behavioural and fMRI session participants were rewarded according to their performance in the game, with a fixed fee of £6 and £8 respectively and an additional amount of money based on their task performance of up to additional £9. At the end of the fMRI sessions, participants were asked to describe what their strategy was in the different social context. Their response revealed a good understanding of the social implication of their choices (Supplementary Table 4). Both in the behavioural and fMRI sessions, the order of the condition was kept constant (cooperation-competition-intermediate) as we wanted all couples to have the same history of interactions.

Visual stimuli were generated from client computers using Presentation software (Neurobehavioral Systems) controlled by a common server running the master script in MATLAB. The stimuli were presented to the players simultaneously. Each experiment was preceded by a short tutorial where players could experience a few trials in each of the three sessions to allow probing the effect of the variability in the task parameter.

## Payoff matrix

We computed a payoff matrix for the Space Dilemma in the following way. Since the target position in each trial is random, the reward in each trial will also be random, but because the target position is sampled from a uniform distribution, each position in the space is associated with an expected payoff which depends on the position of the other player (Fig. 1b). In a two-player game, the midpoint maximizes the chance of winning the trial. For simplicity we therefore assume that players can either compete, positioning in the middle of the space and maximizing their chance of winning, or cooperate, deviating from this position by a distance $\Delta$ to sample the space and maximize the dyad's reward. For all combinations of competitive and cooperative choice, we can build an expected (average) payoff matrix which depends parametrically on $\Delta$. We defined $R$ as the expected reward for each of two players cooperating with each other, $T$ as the expected temptation payoff for someone who decides to compete against a player who is cooperating. $S$ is the "sucker" payoff for a cooperator betrayed by its partner. $P$ is the punishment payoff when both players compete all the times. $R$, $T$, $S$ and $P$ can be computed analytically integrating over all possible position of the target and are equal to:

$$R = \left(\frac{3}{8} + \frac{\triangle}{2} - \triangle^2\right) \qquad (3)$$

$$T = \alpha\left(\frac{3}{8} + \frac{\triangle}{2} - \frac{\triangle^2}{8}\right) + (1-\alpha)\left(\frac{3}{8} - \frac{5\triangle^2}{8}\right) \qquad (4)$$

$$S = \alpha\left(\frac{3}{8} - \frac{5\triangle^2}{8}\right) + (1-\alpha)\left(\frac{3}{8} + \frac{\triangle}{2} - \frac{\triangle^2}{8}\right) \qquad (5)$$

$$P = \frac{3}{8} \qquad (6)$$

The expected reward for cooperative players R is the same in all conditions. This is because the expected reward is equal to the average of the possible rewards associated with win and loss and players who cooperate with equal $\Delta$ have an equal chance of winning the trial.

Therefore $R = (R_{win} + R_{lose})/2 = (\alpha R_{trial} + (1-\alpha)R_{trial})/2 = R_{trial}/2$ which does not depend on $\alpha$. Likewise for the expected reward for competitive players $P$. When one player cooperates and the other competes however, players don't have the same chance of winning a trial and therefore $T$ and $S$ depend also on $\alpha$. For $\alpha = 0.5$ the reward is shared equally no matter what players do so if one compete against a cooperator, they both are expected to win:

$$T = S = \frac{3}{8} + \frac{\triangle}{4} - \frac{3\triangle^2}{8} \qquad (7)$$

For $\alpha = 2$, $T$ diverges quickly from $S$ as

$$T - S = \frac{3}{2}\left(\triangle + \triangle^2\right) \qquad (8)$$

We also computed the expected payoff by simulating 10000 trials of two players competing and/or cooperating by $\Delta$ in the three conditions of the game and the results matched the analytical solutions. For the intermediate and competitive conditions, for all values of $\Delta$ it is also true that $(T > R > P > S)$ thus demonstrating that the Space Dilemma in these conditions is a continuous probabilistic form of Prisoner's Dilemma in the strong sense. For $\Delta > 0.4$ and in all conditions the payoff for a dyad always cooperating is always higher that for one where one player is always competing and other always cooperating or if both alternate cooperation and competition ($2R > T + S$), therefore for $\Delta > 0.4$ the space dilemma is a probabilistic form of iterated prisoner's dilemma. Furthermore, for all conditions the maximum payoff for the dyad is reached for $\Delta = 0.25$.

**Modelling**

To model the behaviour in the game we fitted eighteen different models belonging to three different classes all assuming that players implement some sort of "titxtat". The first class of models (Model S1-S4) is based on the assumption that players decide their behaviour simply based on the last observed behaviour of their counterpart, by reciprocating either their last position, their last change in position, or a combination of the two. A second class of models goes further in assuming that a player learns to anticipate the co-player's position in a fashion that is predicted quantitatively by a Bayesian learner ("Bayesian models" in B1-B8). The eight Bayesian models differ in how this expectation is mapped into a choice, allowing for different degrees of influence of the context, their counterpart behaviour and the player own bias. A third class of models assumes that participants were choosing what to do based not only on the other player behaviour but also on the outcome of each trial, with different assumptions on how winning a trial should change their behaviour in the next (becoming more or less cooperative). This class of models were effectively assuming that the player behaviour would be shaped by the reward collected ("Reward models" in Fig. 3d).

For simplicity, we remapped positions in the space to a cooperation space so that choosing the midpoint (competitive position) would correspond to minimum cooperation while going to the extreme ends of the space (either x = 0 or x = 1) would correspond to maximum cooperation. Therefore $\theta$ is symmetrical to the midpoint and is defined as

$$\theta = |x - 0.5|/0.5 \, (S1 - S4, B1 - B8, R1 - R6) \quad (9)$$

All models include a *precision* parameter capturing intrinsic response variability linked to sensory-motor precision of the participant, such that, given each model's prediction about the player's decision, the actual choice will be normally distributed around that prediction with standard deviation equal to the inverse of the *precision* parameter, constrained to be in the range (0:10000).

For models S1-S4, we assumed that participants were simply reacting to their counterpart recent choice. Model S1 simply assumed that players would attempt to reciprocate their co-player's level of cooperation $\theta$. As the model operate in a symmetrical cooperation space this implies matching their expected level of cooperation in the opposite hemifield.

$$choice(t) \sim N\left(\theta(t-1); 1/Precision\right)(S1) \quad (10)$$

Model S2 simply assumed that players would attempt to reciprocate their co-player's updates in their level of cooperation $\theta$ moving from their previous position plus a fixed SocialBias parameter, capturing their "a priori" desired level of cooperation, constrained to be in the range (−1000:1000).

$$choice(t) \sim N\left(SocialBias + choice(t-1) + \triangle\theta(t-1); 1/Precision\right)(S2) \quad (11)$$

Model S3 was identical to model S2 with the only difference of having three different SocialBias parameters, one for each social context. Model S4 simply assumed that players would reciprocate their co-player's last level of cooperation $\theta$ scaled by a TitXtat multiplicative parameter, constrained to be in the range (0:2). If this is bigger than 1, a participant would cooperate more than the counterpart.

$$choice(t) \sim N\left(SocialBias + TitXtat * \theta(t-1); 1/Precision\right)(S4) \quad (12)$$

For models B1-B8, we used a Bayesian decision framework that has been shown to explain how humans learn in social contexts very well[32,99] for modelling how participants made decisions in the task and how the social context (reward distribution) can modulate these decisions. Our ideal Bayesian learner was assumed to update its expectation about the co-player's level of cooperation $\theta$ on a trial by trial basis by observing the position of its counterpart. In our Bayesian framework, knowledge about $\theta$ has two sources: a prior distribution $P(\theta)$ on $\theta$ based initially on the social context and thereafter on past experience and a likelihood function $P(D\,|\,\theta)$ based on the observed position of the counterpart in the last trial. The product of prior and likelihood is the posterior distribution that defines the expectation about the counterpart's position in the next trial:

$$P(\theta(t+1)) = P(\theta(t+1)|D) = \frac{(P(D|\theta(t)) * P(\theta(t)))}{P(D)} \, (B1 - B8) \quad (13)$$

According to Bayesian decision theory (Berger, 1985; O'Reilly et al., 2013), the posterior distribution $P(\theta\,|\,D)$ captures all the information that the participant has about $\theta$. In the first trial of a block, when players have no evidence on past position of the co-players, we chose normal priors that correspond to the social context: in the competition context $\mu_{prior} = 0$, in the cooperation context, $\mu_{prior} = 1$, and in the intermediate context where the winner takes all, $\mu_{prior} = 0.5$, whereas in all cases the standard deviation is fixed to $\sigma_{prior} = 0.05$ which heuristically speeds up the fit. The likelihood function is also assumed to be a normal distribution centred on the observed location of the co-player with standard deviation fixed to the average variability in positions observed so far in the block (that is, in all trials up to the one in which is estimated). Being the product of two Gaussian distribution the posterior distribution is also Gaussian. All distributions are computed for all values of the linear space at a resolution of $d\theta = 0.01$.

While all Bayesian models assume that players update their expectations about the co-player choices, they differ in how they translate these expectations into their own choices. We built 8 Bayesian models based on increasing level of complexity. In short, all models include a *Precision* parameter. Model B1 simply assumes that players will aim to reciprocate the expected position of the co-player (*coplayer_exp_pos*).

$$coplayer\_exp\_pos(t) = E(P(\theta(t)))(B1 - B8) \quad (14)$$

$$choice(t) \sim N\left(coplayer\_exp\_pos(t); 1/Precision\right)(B1) \quad (15)$$

Model B2 assumes that players will aim for a level of cooperation shifted compared to *coplayer_exp_pos*. Such a shift is captured by the SocialBias parameter which sets an "a priori" tendency to be more or less cooperative and all further Bayesian models include it.

$$choice(t) \sim N\left(coplayer\_exp\_pos(t) + SocialBias; 1/Precision\right)(B2) \quad (16)$$

Model B3 further assumes that participants can fluctuate in how much they reciprocate their co-player cooperation. This effect is modelled multiplying *coplayer_exp_pos* by a *TitXTat* parameter.

$$choice(t) \sim N\left(TitXTat * coplayer\_exp\_pos(t) + SocialBias; 1/Precision\right)(B3) \quad (17)$$

Model B4 further assumes that players keep track of the target position, updating their expectations after each trial in a similar way as they keep track of the co-player position, with a Bayesian update. They then decide their level of cooperation based on the prediction of Model 3 plus a linear term that depends on the expected position of the target scaled by a TargetBias parameter. As the target was random we did not expect this model to significantly increase the fit compared

to Model 3.

$$choice(t) \sim N\left(TitXTat * coplayer\_exp\_pos(t) + SocialBias + TargetBias * \left(P\left(x_{target}\right)\right); 1/Precision\right) (B4) \quad (18)$$

Model B5 further assumes that participants modulate how much they are willing to reciprocate their co-player behaviour based on the social risk associated to the context. In this model the TitXtat takes the form of a multiplying TitXTat factor

$$TitXTat\ factor = \frac{1}{1 + q\_risk * social\_risk} (B5) \quad (19)$$

$$choice(t) \sim N\left(TitXTat\ factor * coplayer\_exp\_pos(t) + SocialBias + TargetBias * \left(P\left(x_{target}\right)\right); 1/Precision\right)(B5) \quad (20)$$

Where $q\_risk$ is a parameter capturing the sensitivity to the social risk induced by the context, which is proportional to the redistribution parameter $\alpha$:

$$social\ risk = 2\,\alpha - 1\,(B5 - B8) \quad (21)$$

Model B6, B7 and B8 do not include the target term. They all model the TitXtat factor with two parameters as in

$$TitXTat\ factor = \frac{TitXTat}{1 + q\_risk * social\_risk} (B6 - B8) \quad (22)$$

$$choice(t) \sim N\left(TitXTat\ factor * coplayer\_exp\_pos(t); 1/Precision\right)(B6 - B8) \quad (23)$$

Model B7 and B8 further assume that participants estimate the probability that their co-player will betray their expectations and behave more competitively than expected. This is computed updating their betrayal expectations after each trial in a Bayesian fashion using the difference between the observed and expected position of the co-player to update a distribution over all possible discrepancies. This produces, for each trial, an expected level of change in the co-player position. Model B7 and B8 both weigh this 'expected betrayal' with a betrayal sensitivity parameter and add this 'betrayal term' either to the social risk, increasing it by an amount proportional to the expected betrayal (model B7) or to the choice prediction, shifting it towards competition by an amount proportional to the expected betrayal (model B8). Model B6 does not include any modelling of the betrayal.

For models R1-R6, we assumed that participants were simply adjusting their position based on the feedback received in the previous trial. Model R1 assumed that after losing, players would become more competitive and after winning, more cooperative. These updates in different directions would be captured by two parameters $Shift_{win}$ and $Shift_{lose}$ both constrained to be in the range (0:10).

$$choice(t) \sim N(choice(t-1) \pm Shift_{(win,lose)}; 1/Precision) (R1) \quad (24)$$

Model R2 assumed that after losing, players would shift their position in the opposite direction than they did in the previous trial, while after winning, they would keep shifting in the same direction. These updates in different directions would be captured by two parameters $Shift_{win}$ and $Shift_{lose}$ both constrained to be in the range (0:10).

$$choice(t) \sim N(choice(t-1) \pm Shift_{(win,lose,sign(\triangle choice(t-1))}; 1/Precision) (R2) \quad (25)$$

Model R3 and R4 are similar to model R1 and R2 in how they update the position following winning or losing but now players would also take into account their co-players last level of cooperation $\theta$ scaled by a TitXtat multiplicative parameter and their own "a priori" tendency to be more or less cooperative captured by a SocialBias parameter.

$$choice(t) \sim N(SocialBias + TitXTat * \theta(t-1) \pm Shift_{(win,lose)}; 1/Precision) (R3) \quad (26)$$

$$choice(t) \sim N(SocialBias + TitXTat * \theta(t-1) \pm Shift_{(win,lose,sign(\triangle choice(t-1))}; 1/Precision) (R4) \quad (27)$$

Model R5 and R6 are identical to model R1 and R2 with the only difference of fitting each choice using the actual value of the previous choice made by the players rather than its fitted value (to prevent under fitting because of recursive errors).

We fit all models to individual participant's data from all three social contexts using custom scripts in MATLAB and the MATLAB function *fmincon*. Log likelihood was computed for each model by

$$LL(model) = \sum_{subjects} \sum_{t} LL(choice(t)) \quad (28)$$

where

$$LL(choice(t)) = \log\left(\sqrt{\frac{Precision}{2\pi}} * \exp\left(-0.5 * ((choice(t) - prediction(t)) * Precision)^2\right.\right) \quad (29)$$

We compared models computing the Bayesian information Criterion

$$BIC(model) = k \log(n) - 2 * LL(model) \quad (30)$$

where $k$ is the number of parameters for each model and $n$ = number of trials * number of participants.

All Bayesian models significantly outperformed both the simple reactive models and the rewards-based ones. To validate this modelling approach and confirm that players were trying to predict others' positions rather than just reciprocating preceding choices, we ran a regressions model to explain participants' choices based on both the last position of the co-player and its Bayesian expectation in the following trial (see supplementary figure 6b).

The winning model is B6, a Bayesian model that contained features that accounted for both people's biases towards cooperativeness, how the behaviour of the other player influenced subsequent choices and the influence of the social context. For this model, participants choose where to position themselves in each trial based on (21), (22) and (23).

*Precision, SocialBias, TitXTat, q\_risk* are the four free parameters of the model. Notice that *TitXTat* is a parameter capturing the context-independent amount of titxtat which is then normalised by the context-dependant social risk.

## Model parameter recovery analysis

We assessed the degree to which we could reliably estimate model parameters given our fitting procedure. More specifically, we generated one simulated behavioral data set (i.e., choices for an interacting couple for 60 trials in three different social contexts) using the average parameters estimated originally on the real behavioral data. Additionally we generated five more simulated behavioral data sets using five randomly sampled parameter sets from the range used in the original fit. For each simulated behavioral data set we ran the winning model B6 this time trying to fit the generated data and identify the set of model parameters

that maximized the log-likelihood in the same way we did for original behavioral data. To assess the recoverability of our parameters we repeated this procedure 10 times for each simulated data set (i.e., 60 repetitions). The recoverability of the parameters was high in almost all cases as can be seen in Supplementary Fig. 6c.

## Model-based regressors

The Bayesian framework allowed us to derive how counterparts' position influenced participants' initial impressions of the level of cooperation needed in a given context. Given this framework, we measured how much the posterior distribution over the co-player position differs from the prior distribution. We did so by computing, for each trial, the Kullback–Leibler divergence (KLD) between the posterior and prior probability distribution over the co-player response. This absolute difference formally represents the degree with which P2 violated P1's expectation and is a trial-by-trial measure of a "social prediction error" that triggers a change in P1's belief, guiding future decisions. A greater KL divergence indicates a higher cooperation-competition update. We, therefore, estimated a social prediction error signal by computing the surprise each player experienced when observing the co-player position, based on its current expectation. In the following equation, where $p$ and $q$ represent respectively prior and posterior density functions over the co-player position, the KL divergence is given by:

$$KLD(p, q) = -\int p(x) \log q(x)dx + \int p(x) \log p(x)dx = \int p(x)(\log(p(x) - \log q(x))dx$$
(31)

$KLD$ is vital in our fMRI investigation as it provides an integrated measure of the trial-by-trial change that accounts for both the uncertainty about the social context and the dynamic of the opponent. As the KL divergence measures a distance between distributions, it is by definition non negative. Therefore it does not provide information about the direction of change between the distributions. We can think of it as an unsigned prediction error capturing the strength of the update. To capture the direction of change we also compute its sign by

$$KLDsign(p, q) = \begin{cases} 1 \ if \int x \, q(x)dx > \int x \, p(x)dx \\ -1 \ otherwise \end{cases}$$
(32)

Therefore we consider a positive $KLDsign$ if the co-player is more cooperative than expected and therefore, after observing the co-player behaviour, the co-player is expected to be more cooperative in the next trial.

These estimates are fundamental to identify the brain areas that covary with the extent and the directionality with which participants update their expectation about their counterparts' strategies given the social context. For this, $KLD$ and $KLDsign$ were used as a parametric regressors in the fMRI analysis.

## MRI data collection

We acquired the fMRI data using a 3T Philips Achieva MRI scanner (Philips, Netherlands). Specifically, we collected functional Echo-Planar-Imaging (EPI) data using a 32-channel SENSE head coil with an anterior–posterior fold over direction (SENSE factor: 2.3; repetition time: 1.5 s; echo time: 40 ms; number of slices: 40; number of voxels: 68 × 68; in-plane resolution: 3 × 3 mm; slice thickness: 3 mm; flip angle: 80°). Slices were collected in an interleaved order. Altogether, we collected three separate runs of 450 volumes each, corresponding to three blocks of 60 trials each for a total of 180 trials in the main experimental task. A pair of participant saw their first scan interrupted after 193 volumes (30 trials) for a technical problem. Another pair of participant was run with a TR of 3 s by mistake. Anatomical images were acquired using a MPRAGE T1-weighted sequence that yielded images with a

$1 \times 1 \times 1$ mm resolution (160 slices; number of voxels: 256 × 256; repetition time: 8.2 ms; echo time: 3.7 ms). We also acquired a B0 map using a multi-shot gradient echo sequence which was subsequently used to correct for distortions in the EPI data due to B0 inhomogeneities (echo time: 2.3 ms; delta echo time: 5 ms; isotropic resolution: 3 mm; matrix: 68 × 68 × 32; repetition time: 383 ms; flip angle: 90°).

## fMRI pre-processing

These volumes were used for the statistical analysis presented in this study. Pre-processing of our data was performed using the FMRIB's Software Library (Functional MRI of the Brain, Oxford, UK) and included: head-related motion correction, slice-timing correction, high-pass filtering (>100 s), and spatial smoothing (with a Gaussian kernel of 8 mm full-width at half maximum). To register our EPI image to standard space, we first transformed the EPI images into each individual's high-resolution space with a linear six-parameter rigid body transformation. We then registered the image to standard space (Montreal Neurological Institute, MNI) using FMRIB's Non-linear Image Registration Tool with a resolution warp of 10 mm.

## fMRI analyses

We performed whole-brain statistical analyses of functional data using a multilevel approach within the generalized linear model (GLM) framework, as implemented in FSL through the FEAT module:

$$Y = X\beta + \varepsilon = \beta_1 X_1 + \beta_2 X_2 + \ldots + \beta_N X_N + \varepsilon$$
(33)

where $Y$ is a $T \times 1$ ($T$ time samples) column vector containing the times series data for a given voxel, and $X$ is a $T \times N$ ($N$ regressors) design matrix with columns representing each of the psychological regressors convolved with a hemodynamic response function specific for human brains[100,101]. $\beta$ is a $N \times 1$ column vector of regression coefficients and $\varepsilon$ a $T \times 1$ column vector of residual error terms. Using this framework we initially performed a first-level fixed effects analysis to process each individual experimental run which were then combined in a second-level mixed-effects analysis (FLAME 1 + 2) treating session as a random effect, and a third level to combine data across subjects, treating participants as a random effect. (We had the same number of sessions across participants). For all analysis, we performed a cluster inference using a cluster-defining threshold of $|Z| > 3.1$ with a FWE-corrected threshold of $P = 0.001$. Time series statistical analysis was carried out using FMRIB's improved linear model with local autocorrelation correction. Applying this framework, we performed the GLMs highlighted below.

**GLM 1**. Our first GLM model included four unmodulated stick regressors aligned with (i) the beginning of the trial (TRIAL in Supplementary Table 2) (ii) the player response (PR) (iii) the time at which the response of its opponent was revealed (OR) (iv) the time at which the target appeared (TARGET). Additionally, we included six regressors capturing trial-by-trial specific information: (1) a stick function at (i) the beginning of the trial parametrically modulated by the expected position of the co-player as derived through the prior distribution for that trial obtained from the Bayesian model (PriorPos). (2) a stick function at (ii) response time modulated by trial by trial changes in the level of cooperation chosen by the player (Pcoop). (3 and 4) two stick functions at (iii) the time at which the response of the co-player was revealed parametrically modulated respectively by the value of the KL divergence between prior and posterior computed in that trial (absPE) and its sign (signPE). The latter could only take the value +1 and −1. Finally (5 and 6) two stick functions at (iv) the time at which the target appeared parametrically modulated respectively by the value of the reward allocated in the trial (Rew) and one signalling whether the player won or lose (Win). The latter could only take the value +1 and −1. All parametrically modulated regressors were z-scored.

**GLM 2**. Our second GLM model included a single boxcar covering the duration of the trial from its onset to the target appearance.

For both GLM we looked both at the average across the three contexts and the contrast between the competitive and cooperative context.

**GLM 3**. Our third GLM model was identical to GLM 1 in all respects but the regressors encoding the prediction error. Here, in order to capture the full parametric effect of the PE instead of having two parametrical regressors we had four unmodulated regressors for four different trial groupings based on the KLD value and its sign. In short, we binned trials in four group based on their absPE and sign PE value (high positive, low positive, low negative and high negative values). The cut-off value to distinguish high and low prediction errors was set to be the median value across all prediction errors with the same sign. Each of the four regressors was an unmodulated stick regressor aligned with the time at which the response of the co-player was revealed in trials belonging to the corresponding bin.

**ROI analysis**. To quantify the modulation of the activity across conditions, we extracted the average signal of the neural activation for all three social contexts in regions of interest (ROIs), defined as either three or five-voxel radius spherical masks placed centred on the peak of the activations at the group level. We back projected this masks and extracted individual participant betas. We split each participant's time series into trials resampled each trial to 10 s at a resampling resolution of 50 ms. We then carried out a general linear model across trials at every time point in each participant independently. Lastly, we calculated group average effect sizes at each time point, and their standard errors. To analyse the predictive power of an area, we split trials in two groups based on whether in the next trial the player was more cooperative or more competitive. For each of the two groups, we extracted the time-courses of the BOLD signal in the selected ROI at the time of the other player response and examined whether signals on a trial were predictive of a change in behaviour (an increase or decrease in competitiveness) on trial ($t + 1$). To test the full parametric effect of the prediction error in the two in clusters in TPJ we computed the average population betas within the two ROIs for each of the four PE regressors of GLM 3, corresponding to four group of trials based on their absPE and sign PE value (high positive, low positive, low negative and high negative values).

### Reporting summary

Further information on research design is available in the Nature Portfolio Reporting Summary linked to this article.

## Data availability

The pre-processed fmri and behavioural data generated in this study have been deposited in an Open Science Framework project [https://osf.io/sydea]. The raw fMRI data are protected and are not available due to data privacy laws.

## Code availability

The code to generate the results and the figures of this study is available in an Open Science Framework project [https://osf.io/sydea].

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

## Acknowledgements

This work was supported by the Economic and Social Research Council (ESRC; grant ES/L012995/1 to M.G.P.), the Biotechnology and Biological Sciences Research Council (BBSRC; David Phillips Fellowship BB/R010668/2 to M.A.J.A.) and by the UK Research and Innovation (UKRI; grant MR/T023007/1 to E.F.F.). We also thank Frances Crabbe for assistance with data collection.

## Author contributions

M.A.P., E.F.F. and M.G.P. designed the experiments. M.A.P. and D.H.A. collected the data. M.A.P., E.F.F., M.A.J.A. and M.G.P. analysed the data and wrote the paper. All authors discussed the results and implications and commented on the manuscript at all stages.

## Competing interests

The authors declare no competing interests.
