## [Peer Review File · Nature Communications]

Neural implementation of computational mechanisms underlying the continuous trade-off between cooperation and competitionReviewers' comments:

Reviewer #1 (Remarks to the Author):

This study investigated how cooperation and competition, as conceptualized on a continuous dimension following Bayesian principles, are implemented at the neural level. The authors developed a novel task to assess cooperation and competition on a continuous dimension. By modulating the social context from cooperative to competitive, this task allowed an investigation of social prediction update and the change of social strategy along the cooperation-competition dimension. They found that the social context as operationalized by the game partner's intention changed the participant's behavior in a Bayesian manner through computational modeling. Signed and unsigned Kullback-Liebler divergence (a form of Bayesian surprise) were distinctively correlated with anterior and posterior TPJ.

I think the design of the task is smart and the task has considerable potential for future studies in decision making in the context of cooperation and competition. Both behavioral and neuroimaging results are intriguing and have potential impact in the field as well as corroborating previous findings. However, there are issues with the interpretation of the results, especially regarding the computational model. Moreover, there is scope for improvement also in other parts.

Major points

- 1) Although the authors portray participant behavior as predicting the other player's behavior, there appears to be no evidence for that statement. A tit-for-tat strategy seems compatible with reactive behavior. Can you provide any behavioral evidence for predictions?
- 2) The model the authors suggested seems to work well to explain the behavior. However, the evidence to support model selection seems not strong enough. The authors tested eight models in total and figure 3d shows the model comparison results. From this figure, the difference among models 6-8 is very small and models 7 and 8 seem to have even lower BIC or at least very similar BIC as model 6. This makes it difficult to understand why the authors chose model 6 as best model and the issue remains unexplained in the manuscript. Thus, we are left with the impression that the authors arbitrarily selected the wrong model. Including the model comparison results as a table and providing more convincing evidence to support model 6 compared to other models is necessary.
- 3) The manuscript does not describe how the computational modeling, simulation and parameter recovery were done. Details on these aspects are critical to understand the computational modeling procedure as well as to ensure reproducibility and open science. Please include information such as the software used for the modeling, number of iterations, the setting for the model estimation, how the best model was determined, how the data for simulation was generated, how many simulation was done, et cetera. There must be sufficient information for people to reproduce the computational modeling part.
- 4) Given that prior experience changes behavior in the game, the order must have an effect on Bayesian updating and behavior. Thus, the order of the conditions should be considered in the experimental design. As the order of cooperative-intermediate-competitive condition seems to have been fixed, could you provide rationale for this? In a similar line, participants experienced some trials of each condition before the experiment. Was that accounted for in the model? Or how did you ensure that the practice trials did not affect the prior of participants?
- 5) The sample size for fMRI data that actually used for analysis is rather small considering the typical number of participants in recent fMRI studies without applying strict criteria based on power analysis (https://www.frontiersin.org/articles/10.3389/fnhum.2018.00016/full?utm_source=F-AAE&utm_medium=EMLF&utm_campaign=MRK_527044_55_Neuros_20180130_arts_A). Do you have any rationale for this?
- 6) Providing all neuroimaging results as a table with cluster size would be helpful.

Minor points

- 1) Line 141: How was the winner/loser determined if the two players were equidistant from the target and was this considered when determining optimal behavior/calculating the payoff matrix?

- 2) Figure 1a: Please provide durations for all frames, not just the first (for the frames under participant control provide median, minimum-maximum duration) and add intertrial interval frame with its duration.
- 3) Some characters in the caption of Figure s1 are broken (in the separate file, not the one in the manuscript). Please check and replace.
- 4) Figure S2: the legend states that players positioned themselves on average in the middle of the hemifields (around 0.3 away from the midpoint). However, Fig. S2a shows that they positioned themselves at 0.2 on average, in contrast to the legend. Moreover, Fig. S1a indicates that the middle of the hemifield is at 0.25 from the midpoint, providing yet another value. These (apparent?) conflicts should be resolved.
In addition, Figure S2d appears to show that one pair was more extreme in the competition condition than the other pairs. How did they differ in their behavior from the rest?
- 5) Line 183 mentions that the best long term strategy for the dyad was to always cooperate regardless of condition. However, it is unclear why this should matter for the individual? Moreover, the statement seems to assume infinite play and ignore end-game effects, which obviously is unrealistic.
- 6) In supplement fig3b, correlation r is missing.
- 7) The equation for punishment P seems to be missing on page 28.
- 8) What are the grey lines in figure 3e? Unlike for the grey and red dots, the meaning of the grey lines was not indicated in the caption. It needs to be explained as well.
- 9) In Fig3g, the y and x axis labels are confusing. Adding 'precision' next to it would facilitate understanding. In addition, I'm curious why the correlation in Fig3g is beta instead of r which was used in Fig3e and Fig3f.
- 10) Line 450: condition B is unclear.
- 11) Line 996: The authors use a binary term to approximate signed prediction errors. However, this discrete measure is not the same as the continuous notion of a proper prediction error and the corresponding parametric modulator seems more appropriately viewed as cooperation vs. competition contrast.
- 12) On page 33, the authors used monkey rather than human HRF. What is the rationale for this? Do the findings stay the same if you use human HRF instead?
- 13) There are many mistakes in the writing, such as typos, incomplete/wrong sentence structures, inconsistent use of brackets, wrong capitalizations, or misuse of abbreviation throughout the manuscript. Please check carefully.
(e.g. line 420-425, 429, 506, 819, 868, 1053, 1208, figure s1, C for cooperation, (D) for defect cooperation)

Reviewer #2 (Remarks to the Author):

This manuscript tests neural computations involved in cooperative and competitive decisions. In an fMRI experiment, participants chose to cooperate more or less by moving along a one-dimensional space. Unsigned and signed prediction errors related to a partner's cooperation were observed in distinct portions of rTPJ, whereas other brain regions encoded prediction errors (e.g., accG) or cooperation (pDMPFC) differently as a function of context. Responses in these regions also predict shifts in cooperative behavior.

A major strength of this paper is that it brings together a number of factors relevant to cooperation together. As the authors note, previous work has examined factors included in the present work (individual differences, reciprocity, cost), but the present manuscript brings them together in one model, allowing a person x situation interaction approach to cooperation in the brain. The findings are also quite interesting. As a result, I think this paper has the potential to make a very nice contribution to the social learning literature.

That said, I have a few comments that I hope will be helpful for clarifying the theoretical contribution, addressing relevant background literature, and clarifying some of the the methods.

1) The theoretical motivation is currently stated fairly broadly: "We are still lacking an integrated understanding of what factors govern the behavioral and neural trade-off between cooperation and competition." As the next paragraphs go on to detail, a great deal of research has examined what factors govern the tradeoff between cooperation and competition. It would be helpful to outline the more specific contribution of the present work here.

1) There is some past literature that is relevant:

a. Although the game used in this experiment is novel, past behavioral research and theory has indeed examined continuous Prisoner's Dilemmas (Barclay & Willer, 2007; Kilingback, Doebeli, & Knowlton, 1999) including the impact of changing the costs/benefits of cooperation (Capraro, Jordan, & Rand, 2014).

b. Past work has argued for a theoretical view that rTPJ encodes prediction errors (Koster-Hale & Saxe, 2013, *Neuron*), and there is prior evidence for both unsigned prediction errors/surprise (Boorman, O'Doherty, Adolphs, & Rangel, 2013; Dungan, Stepanovic, & Young, 2016; Kim, Mende-Siedlecki, Anzelotti, & Young, 2021; Park, Delgado, Farreri, & Young, 2021) and signed prediction errors (Hackel, Doll, & Amodio, 2015). The present work is the first study I know of to identify both simultaneously in distinct portions of rTPJ—which is quite interesting—but there is reason from past work to think that both types of prediction errors involve this region. There is similarly work linking accG to switches in strategy between cooperation/self-interest (Park, Sestito, Boorman, & Dreher, 2020) and linking rTPJ to positive expectations about others' cooperation (Park, Sestito, Boorman, & Dreher, 2019; Hackel, Wills, & Van Bavel, 2020).

2) Although behavioral research has examined continuous cooperation, it is true that less work has examined continuous cooperation in the brain. However, I think the authors could say more about why this is important. For instance, imagine the present experiment had offered binary choices, in which subjects could only choose the cooperative and competitive positions. Presumably, subjects would have experienced prediction errors related to a partner's cooperation and would have gradually adjusted their probability of cooperating (even if choices are binary). Can the authors say more about the unique insights offered by continuous choices?

3) For the fMRI analysis of "self cooperation" across contexts, the main text says the regressor was "change in cooperation" but the methods say that "level of cooperation" was used. I was unsure which of these was used in the end. If it is level of cooperation, the results appear similar to those observed in response to "negative value signals" in past work, i.e., the value of unchosen versus chosen options (Doll, Duncan Simon, Shohamy, & Daw, 2015). Do responses here predict individual differences in cooperation across these contexts, and/or are they explained by the reduced value of cooperation in the competition context (e.g., cooperative deviations from model predictions for choice)? This might fit with work finding responses in regions linked to value-based choice that reflect variability in cooperation and strategy shifts (Park, Sestito, Boorman, & Dreher, 2019).

4) More generally, the findings regarding pDMPFC and accG could be more clearly described and discussed in the discussion section; currently, the specific findings and their importance are a bit vague.

5) The methods section indicates that the prediction errors variables in the fMRI analyses included (i) the KL divergence and (ii) its sign, which was either 1 or -1. I was unclear on why the signed prediction error was restricted to 1 and -1, rather than examining a scaled parametric modulator (i.e. KL divergence x the sign).

6) Can the authors comment on how well subjects understood the extent to which different choices would be cooperative or competitive?

Minor points:

1) In describing the three factors shaping cooperation in the introduction, the third factor ("how dyads interact with each other") can be broadened to an array of social consequences, which can include both reciprocity in repeated interactions (Dal Bo, Dal Bo & Frechette, 2011) and reputation (Barclay & Willer, 2007; Feinberg, Willer, & Schultz, 2014; see Kraft-Todd et al., 2013, for further discussion)

Reviewer #3 (Remarks to the Author):

This manuscript describes the results of an fMRI study in which participants engage in an extended form of the prisoner's dilemma game aimed at exploring neural computations underlying the capacity to co-operate or compete with an opponent. The authors describe behavioral evidence that participants are sensitive to contextual manipulations that change the benefit of competing, in that they choose to compete less in contexts designed to promote cooperation and more in contexts designed to promote competition. A Bayesian model is used to account for participants trial-by-trial behavior. Various models are compared with different levels of complexity, ranging from merely copying the opponent's position, adding in a bias toward cooperation, incorporating mechanisms for tit for tat and so on. When testing for updates about predictions concerning the expected level of cooperation, activity was reported in posterior STS correlating with the ML divergence of the posterior and prior, describe the amount of update occurring in the predictions – both a signed and unsigned version of this signal was found in separate areas of STS. Furthermore, activity in mPFC correlated with other components of the model, and in which activity was particularly dependent on changes in the social context.

By using a novel task to examine cooperative/competitive decisions the authors have generalized these findings to a more continuous setting as well as replicated previous results about a contribution of TPJ in encoding social prediction errors. The report of different areas of TPJ correlating with signed and unsigned prediction errors adds further insight to the functional properties of this area in social prediction error coding. However, the overall sense from this manuscript is that the findings do not necessarily represent a major advance in our understanding of how the brain mediates decisions about cooperating or competing beyond what is already known about the role of TPJ and dmPFC in social computations. A number of additional control conditions would seem to be needed to make a claim that the authors have identified specific evidence for the neural correlates of cooperation/competition. Furthermore, it is not clear that the authors have clear evidence for Bayesian inference above and beyond other possible model implementations. In addition, a number of concerns are noted about the modeling – which can potentially be remedied with additional analyses.

(1) It is not clear from this manuscript that the authors' reported findings are specifically related to co-operation and competition decisions as opposed to corresponding to social decision-making more generally. The signal the authors report in pSTS corresponds to an update related to a difference between expected and observed behavior of the other agent. In the model and task used, this is couched in terms of cooperation vs competition, but such a signal might not be specific to such a circumstance at all, but simply to any situation where an agent needs to make a prediction about another's behavior and update that prediction. Although differences in signal in mPFC were reported as a function of different contexts, such a manipulation changed the riskiness and potential cost of any decision beyond any specific effects on the participant's likelihood to cooperate vs compete. The addition of control conditions such as a non-social condition matched for the gain/loss profile of a decision, but without a social element, or a control condition with a social element but no cooperative/competitive component could help address some of those concerns about specificity.

(2) The authors make a strong case for the Bayesian model that they use – essentially arguing that their data provides evidence that humans use a Bayesian strategy (or Bayesian principles as they put it) to solve the cooperation/competition problem. The word Bayesian is echoed in the title. However, the authors do not compare their Bayesian strategy to a non-Bayesian one, thus we cannot ascertain whether or not a Bayesian strategy is necessarily the best way to explain participants behavior. At the simplest, one could also imagine some sort of RL mechanism in which current behavior is adjusted based on past feedback (whether one won or not on the previous trial for pursuing a particular

option). One could also imagine more sophisticated non-Bayesian mechanisms also tracking the opponent's strategies and implementing a choice accordingly. There is nothing in principle wrong with the Bayesian modeling approach the authors used— it is convenient and often useful to deploy such a modeling approach— but it is a strong claim for the authors to contend that they have evidence participants specifically use a Bayesian strategy and that they necessarily reveal neural correlates thereof — when other non-Bayesian models could potentially explain the results, especially if constructed to perform some form of social inference/learning about others' actions akin to that which is performed by their Bayesian model.

(3) On a related point — the relationship between the model predictions and human behavior is unclear. While plots of model fits are provided for each subject, it is hard from these plots to gauge how well the models are doing in capturing trial by trial behavior. Posterior predictive checks are needed to ascertain to what extent the various models utilized can capture key behavioral patterns in the human choice data. It would also be important to implement a model simulation parameter recovery analysis to determine to what extent it is possible to uniquely identify each of the parameters included in their model during the model fitting process. Model confusability analyses would also help to identify which models actually make unique behavioral predictions.

(4) It is rather unclear what the mPFC results actually signify in terms of adding to our understanding about how cooperative strategies are implemented at the computational or neural level. For instance, what is the functional significance of a neural signal in dmPFC that shows different signs of social prediction error as a function of different social contexts? What does this signal tell us about how the brain implements this type of decision-making? The impression is that these findings have emerged from an exploratory analysis and that they are not easily interpretable.

Reviewers' comments:

Reviewer #1 (Remarks to the Author):

This study investigated how cooperation and competition, as conceptualized on a continuous dimension following Bayesian principles, are implemented at the neural level. The authors developed a novel task to assess cooperation and competition on a continuous dimension. By modulating the social context from cooperative to competitive, this task allowed an investigation of social prediction update and the change of social strategy along the cooperation-competition dimension. They found that the social context as operationalized by the game partner's intention changed the participant's behavior in a Bayesian manner through computational modelling. Signed and unsigned Kullback-Liebler divergence (a form of Bayesian surprise) were distinctively correlated with anterior and posterior TPJ.

I think the design of the task is smart and the task has considerable potential for future studies in decision making in the context of cooperation and competition. Both behavioral and neuroimaging results are intriguing and have potential impact in the field as well as corroborating previous findings. However, there are issues with the interpretation of the results, especially regarding the computational model. Moreover, there is scope for improvement also in other parts.

We thank the reviewer for highlighting that “the task is smart and has considerable potential for future studies in decision making in the context of cooperation and competition” and for pointing at the potential impact of the data. We also thank you for your constructive comments. We have taken on board all feedback which we feel prompted us to improve the manuscript. We hope we have now address all of the issues highlighted

Major points

1) Although the authors portray participant behavior as predicting the other player's behavior, there appears to be no evidence for that statement. A tit-for-tat strategy seems compatible with reactive behavior. Can you provide any behavioral evidence for predictions?

We thank the reviewer for this suggestion. We agree that a tit-for-tat strategy is compatible with reactive behaviour (being inherently a reactive strategy) and this is reflected in our modelling approach which starts from the assumptions that players reciprocate their co-players position. However, the Bayesian models we employed go beyond a purely reactive strategy in several ways. First, assuming that what is reciprocated is not the position of the co-player in the last trial but rather the expected position (yet unobserved) in the current trial. In this way the Bayesian models take into account the full history of their co-players past behaviour. Furthermore, in our models the extent to which the expected position is reciprocated is modulated by the social context and shifted by a subject specific social bias.

We developed these models precisely because we have evidence that players' behaviour could not be simply explained by the last choices of their co-player or in pure reactive terms.

We have now added this in the paper by first developing a number of purely reactive, non-Bayesian models ("simple" models 1-2-3-4 in the new figure 3) which assume that players only reciprocate either the last position of their counterpart, their last change in position, or a combination of the two. All these models are significantly outperformed by the Bayesian models that fit choices based on expectations of their counterparts' positions. Second, to confirm that players were trying to predict others' positions rather than just reciprocating preceding choices, we ran new regressions models to explain participants' choices based on both the last position of the co-player and its Bayesian expectation in the following trial. We found that expected positions are significantly better predictors than preceding choices (see new supplementary figure 6).

Both these pieces of evidence point to the fact that whilst players implement tit-x-tat, they do so in a way that considers all past behaviour of their co-player, effectively discounting their latest choice with prior decisions. We believe it is not surprising that, after a number of trials, players expect their co-player to behave in a certain way and we think that this can allow their behaviour to be more resilient. For example, if their counterpart deviates suddenly in a more competitive direction from their usual cooperative position, Bayesian participants would discount this sudden change if there was a consistent history of cooperation, thus making their cooperation more robust to single, potentially accidental, deviations. A Bayesian expectation naturally captures this intuitive behaviour. We have now added a paragraph (lines 344-352) to address this point in the text.

2) The model the authors suggested seems to work well to explain the behavior. However, the evidence to support model selection seems not strong enough. The authors tested eight models in total and figure 3d shows the model comparison results. From this figure, the difference among models 6-8 is very small and models 7 and 8 seem to have even lower BIC or at least very similar BIC as model 6. This makes it difficult to understand why the authors chose model 6 as best model and the issue remains unexplained in the manuscript. Thus, we are left with the impression that the authors arbitrarily selected the wrong model. Including the model comparison results as a table and providing more convincing evidence to support model 6 compared to other models is necessary.

We thank the reviewer for their comment. We have now considered this and have added a comparison table (Supplementary Table 1) and extra discussion in the methods. The reviewer is right that the difference in BIC among models B6-8 is small but that is not surprising given that these models are remarkably similar among each other. Models B7 and B8 have one extra parameter compared to model B6 which is modelling the probability that a co-player might "betray" by arbitrarily becoming more competitive. This probability is estimated in a Bayesian fashion based on the history of unexpected deviations. Model B7 differs from model B8 in the way such betrayal probability affects the choice: in B7 it increases the social risk that modulates the titxat factor by multiplying the social context impacts on the titXtat factor, whilst model B8 the betrayal

probability with an additive term. The influence of social context in all three models has the same function: increasing titXtat as the amount of redistribution increases. Aside from modelling the betrayal probability Model B7 and model B8 are identical to model B6. However, the inclusion of the extra parameter is not justified by a small improvement in negative log likelihood ($\text{LL(B8)-LL(B6)}=38$; $\text{LL(B7)-LL(B6)}=7$) due to a significant increase in Bayesian Information Criterion ($\text{BIC(B8)-BIC(B6)}=380$; $\text{BIC(B7)-BIC(B6)}=442$) suggesting that it is unlikely that our players encoded the probability of betrayal independently of the effect of context (which makes subject more cautious anyway). Therefore, while our selection wasn't arbitrary, these models make very similar behavioural predictions, as they are inherently similar since they share the same Bayesian architecture, a significant number of features and three free parameters. Through the additional models included (as outlined in the previous response), we now also provide further evidence for the principles underlying the winning model (i.e. that it needs to be Bayesian) and therefore in favour of model 6.

3) The manuscript does not describe how the computational modelling, simulation and parameter recovery were done. Details on these aspects are critical to understand the computational modelling procedure as well as to ensure reproducibility and open science. Please include information such as the software used for the modelling, number of iterations, the setting for the model estimation, how the best model was determined, how the data for simulation was generated, how many simulation was done, et cetera. There must be sufficient information for people to reproduce the computational modelling part.

We thank the reviewer for their comment that prompted us to significantly expand the methods section related to the model. We now include details about the software, number of iterations, model estimation, parameter recovery analysis and other aspects of the model implementation (see new Modelling section in the methods and the new supplementary figure 6). We also plan to share the code used if the paper is accepted. We hope that the reviewer will feel there is now enough information to reproduce the model.

4) Given that prior experience changes behavior in the game, the order must have an effect on Bayesian updating and behavior. Thus, the order of the conditions should be considered in the experimental design. As the order of cooperative-intermediate-competitive condition seems to have been fixed, could you provide rationale for this? In a similar line, participants experienced some trials of each condition before the experiment. Was that accounted for in the model? Or how did you ensure that the practice trials did not affect the prior of participants?

We thank the reviewer for their comment which allows us to clarify our rationale. We kept the order of the section constant (cooperation, competition, intermediate) precisely because we think the order is likely to have some impact on the behaviour. As our goal was to infer the neural network controlling the competition-cooperation trade off at the population level, we wanted all couples to have the same history of interactions. While it is true that participants came in for a purely behavioural session a few days ahead of the fMRI session, we think this is unlikely to bias their prior in the experiment for two reasons: participants knew they were playing against someone different compared to the behavioural session (we let participants see each other at close distance prior to the start of the sessions and we stressed they would play with

someone different even prior to the behavioural session). Moreover, a significant amount of time passed between the two sessions, ranging from one to three weeks. We now emphasise both reasons in the methods section (lines 952-965).

5) The sample size for fMRI data that actually used for analysis is rather small considering the typical number of participants in recent fMRI studies without applying strict criteria based on power analysis. Do you have any rationale for this?

We thank the reviewer for this observation. Our fMRI sample was indeed half of the collected data (n=50) as people came in pairs. Besides the challenge of collecting interactive data from couples of participants, we do not think that our sample size is insufficient to draw inference about the fMRI activity underlying the behaviour observed. Indeed (Nee 2019, Communications Biology) has shown that high replicability can be achieved for a sample size very similar to ours (n=23 vs ours n=25) for a task duration similar to ours (~1 hour). Furthermore, we made inference at a rather conservative threshold ($p < 0.001$ corrected). We have now added a line in the methods clarifying our sample size choice and referencing this paper.

6) Providing all neuroimaging results as a table with cluster size would be helpful.

We have now added a table (Supplementary Table 2) which summarises all significant neuroimaging activations.

Minor points

1) Line 141: How was the winner/loser determined if the two players were equidistant from the target and was this considered when determining optimal behavior/calculating the payoff matrix?

When participants were equidistant they were both declared winners and the reward was split in two between them. Given the continuous nature of the measure, such trials were rare (<1% trials). We have now added a sentence in the methods to account for this rare case .

2) Figure 1a: Please provide durations for all frames, not just the first (for the frames under participant control provide median, minimum-maximum duration) and add intertrial interval frame with its duration.

Apologies for not including this information, they have now been included.

3) Some characters in the caption of Figure s1 are broken (in the separate file, not the one in the manuscript). Please check and replace.

Fixed

4) Figure S2: the legend states that players positioned themselves on average in the middle of the hemifields (around 0.3 away from the midpoint). However, Fig. S2a shows that they positioned themselves at 0.2 on average, in contrast to the legend. Moreover, Fig. S1a indicates that the middle of the hemifield is at 0.25 from the midpoint, providing yet another value. These (apparent?) conflicts should be resolved. In addition, Figure S2d appears to

show that one pair was more extreme in the competition condition than the other pairs. How did they differ in their behavior from the rest?

We thank the reviewer for these observations, we have now corrected the wrong value - players are indeed 0.2 away from the midpoint. Please also note that whilst in figure S1 position are expressed in absolute values (0-1) with the midpoint being 0.5, in figure S2 the y axis measure the distance from the midpoint, therefore it's symmetric on each of the two hemifields. We adjusted the caption to reflect this. The pair highlighted in figure S2d was rewarded significantly less as they played for less trials (~30) because of a technical glitch that forced their scan in the competitive condition to be interrupted. We have added a sentence in the methods and in the caption to account for this.

5) Line 183 mentions that the best long term strategy for the dyad was to always cooperate regardless of condition. However, it is unclear why this should matter for the individual? Moreover, the statement seems to assume infinite play and ignore end-game effects, which obviously is unrealistic.

Thank you for pointing this out. We merely wanted to say that a purely cooperative strategy has the highest expected return for the couple beating any other strategy. That does not imply that is the best strategy for the individual. We have now added two caveats about our statement on the best long-term strategy which of course requires perfect cooperation for both players and is not to be expected based on end game effects.

6) In supplement fig3b, correlation r is missing.

Corrected.

7) The equation for punishment P seems to be missing on page 28.

Fixed

8) What are the grey lines in figure 3e? Unlike for the grey and red dots, the meaning of the grey lines was not indicated in the caption. It needs to be explained as well.

They are individual subject fits as we now specify in the caption.

9) In Fig3g, the y and x axis labels are confusing. Adding 'precision' next to it would facilitate understanding. In addition, I'm curious why the correlation in Fig3g is beta instead of r which was used in Fig3e and Fig3f.

We added precision to the label and substituted r to beta.

10) Line 450: condition B is unclear.

We were referring to the intermediate condition. We have now corrected it.

11) Line 996: The authors use a binary term to approximate signed prediction errors. However, this discrete measure is not the same as the continuous notion of a proper prediction error

and the corresponding parametric modulator seems more appropriately viewed as cooperation vs. competition contrast.

We thank the reviewer for their point. The sign of the prediction error doesn't contrast cooperation vs competition per se but rather looks at the changes in social orientation by contrasting trials in which cooperation is increased vs trials in which it is decreased compared to the previous trial. We included in our GLM two regressors related to the prediction error, one with the value of the KL divergence (which is by its definition, always positive) representing the surprise associated with the change in the co-player position and one with the direction of the change (encoded through a signed value). Using simultaneously the KL divergence and the signed prediction error produces two perfectly collinear regressors in half the trials. To test the full parametric effect of the different clusters in TPJ we run two control GLMs (see methods): one including only a signed prediction error (signed KLD) and another one with four unmodulated regressors for four different trial groupings based on the KLD value and its sign. Whilst in the first GLM activations in raTPJ remained under threshold, probably due to the fact that the activity encodes prediction errors asymmetrically across the two contexts, in the second control GLM we see a clear modulation of raTPJ activity with value of prediction errors signalling increases in competition of the co-player (see Figure 4d) whilst activations in rpTPJ show a u-shaped relationship with the value of KLD, consistent with our interpretation of a signal reflecting the sign or the absolute value of the prediction error, respectively. We now comment on this additional analysis in the main text (lines 454-460).

12) On page 33, the authors used monkey rather than human HRF. What is the rationale for this? Do the findings stay the same if you use human HRF instead?

We actually used a human HRF. We apologise for the confusion, we have now corrected the mistake.

13) There are many mistakes in the writing, such as typos, incomplete/wrong sentence structures, inconsistent use of brackets, wrong capitalizations, or misuse of abbreviation throughout the manuscript. Please check carefully.

(e.g. line 420-425, 429, 506, 819, 868, 1053, 1208, figure s1, C for cooperation, (D) for defect cooperation)

We apologise for these mistakes, we believe all mistakes are now fixed and typos corrected.

Reviewer #2 (Remarks to the Author):

This manuscript tests neural computations involved in cooperative and competitive decisions. In an fMRI experiment, participants chose to cooperate more or less by moving along a one-dimensional space. Unsigned and signed prediction errors related to a partner's cooperation were observed in distinct portions of rTPJ, whereas other brain regions encoded prediction errors (e.g., accG) or cooperation (pDMPFC) differently as a function of context. Responses in these regions also predict shifts in cooperative behavior.

A major strength of this paper is that it brings together a number of factors relevant to cooperation together. As the authors note, previous work has examined factors included in the present work (individual differences, reciprocity, cost), but the present manuscript brings them together in one model, allowing a person x situation interaction approach to cooperation in the brain. The findings are also quite interesting. As a result, I think this paper has the potential to make a very nice contribution to the social learning literature.

We thank the reviewer for their words and the positive and constructive feedback which prompted us to improve the manuscript.

That said, I have a few comments that I hope will be helpful for clarifying the theoretical contribution, addressing relevant background literature, and clarifying some of the methods.

- 1) The theoretical motivation is currently stated fairly broadly: "We are still lacking an integrated understanding of what factors govern the behavioral and neural trade-off between cooperation and competition." As the next paragraphs go on to detail, a great deal of research has examined what factors govern the tradeoff between cooperation and competition. It would be helpful to outline the more specific contribution of the present work here.

We thank the reviewer for their comment. We have now outlined what we believe is the more specific contribution of our paper already in the introductory remark highlighted where we now say

"We are still lacking an integrated understanding of how the brain controls and arbitrates over the continuous trade-off between cooperation and competition, and

more specifically, which neural mechanisms and computational principles are involved.”

1) There is some past literature that is relevant:

a. Although the game used in this experiment is novel, past behavioral research and theory has indeed examined continuous Prisoner’s Dilemmas (Barclay & Willer, 2007; Kilingback, Doebeli, & Knowlton, 1999) including the impact of changing the costs/benefits of cooperation (Capraro, Jordan, & Rand, 2014).

b. Past work has argued for a theoretical view that rTPJ encodes prediction errors (Koster-Hale & Saxe, 2013, Neuron), and there is prior evidence for both unsigned prediction errors/surprise (Boorman, O’Doherty, Adolphs, & Rangel, 2013; Dungan, Stepanovic, & Young, 2016; Kim, Mende-Siedlecki, Anzelotti, & Young, 2021; Park, Delgado, Farreri, & Young, 2021) and signed prediction errors (Hackel, Doll, & Amodio, 2015). The present work is the first study I know of to identify both simultaneously in distinct portions of rTPJ—which is quite interesting—but there is reason from past work to think that both types of prediction errors involve this region. There is similarly work linking accG to switches in strategy between cooperation/self-interest (Park, Sestito, Boorman, & Dreher, 2020) and linking rTPJ to positive expectations about others’ cooperation (Park, Sestito, Boorman, & Dreher, 2019; Hackel, Wills, & Van Bavel, 2020).

We thank the reviewer, we completely agree with their point that there is pre-existing evidence for prediction errors in rTPJ and other examples of continuous prisoner’s dilemmas have been proposed. We thank them for reminding us of this literature that we now reference in the discussion, where we now highlight how this literature broadly supports our results.

2) Although behavioral research has examined continuous cooperation, it is true that less work has examined continuous cooperation in the brain. However, I think the authors could say more about why this is important. For instance, imagine the present experiment had offered binary choices, in which subjects could only choose the cooperative and competitive positions. Presumably, subjects would have experienced prediction errors related to a partner’s cooperation and would have gradually adjusted their probability of cooperating (even if choices are binary). Can the authors say more about the unique insights offered by continuous choices?

Thank you for this comment that allowed us to clarify our thoughts on this point. We think that a continuous set up allows observing the changes in behaviour that would otherwise remain “latent” in a binary setting if encoded, as suggested, as probability of defecting cooperation. This is important, as in a continuous task, minor adjustments of behaviour are observable by the co-player who can react to it, inducing dynamics that could remain undetected in a binary setting. For instance, in our intermediate condition we see the players slowly drifting towards the more competitive position. Therefore, our task provides direct evidence of the fine tuning of cooperation behaviour and of its neural underpinnings. Moreover, this is important for examining neural activity. In binary choice tasks, the same degree of intention to cooperate might be reflected in identical choices being made for multiple trials before a sudden shift by a co-player revealing a large prediction error. Thus the gradation of prediction error

values that best accounted for shifts in people's behaviour and intentions to compete would not be well captured. We have now added a sentence in the conclusion which makes this point at lines 605-618.

3) For the fMRI analysis of "self cooperation" across contexts, the main text says the regressor was "change in cooperation" but the methods say that "level of cooperation" was used. I was unsure which of these was used in the end. If it is level of cooperation, the results appear similar to those observed in response to "negative value signals" in past work, i.e., the value of unchosen versus chosen options (Doll, Duncan Simon, Shohamy, & Daw, 2015). Do responses here predict individual differences in cooperation across these contexts, and/or are they explained by the reduced value of cooperation in the competition context (e.g., cooperative deviations from model predictions for choice)? This might fit with work finding responses in regions linked to value-based choice that reflect variability in cooperation and strategy shifts (Park, Sestito, Boorman, & Dreher, 2019).

We thank the reviewer for spotting this inconsistency. Indeed, in our fMRI analysis we use changes in cooperation and this is now reflected in the text in the methods. The degree to which the activity that we report in pDMPFC and dACC reflects increases in cooperation does correlate with individual differences in cooperation as measured through the social bias parameter of our model (See supplementary Figure 5e) and its contextual modulation is compatible with the notion that these region encode "out-of-context" behaviour, that is, they positively signal increases in cooperation in competitive context while doing the opposite in the competitive context. While we don't think that the activity in the cingulate that we report strictly reflect a valuation we do think that this "out-of-context" signalling is compatible with a reduced value of cooperation in the competitive context and might fit with the results of (Park, Sestito, Boorman, & Dreher, 2019) which we now cite.

4) More generally, the findings regarding pDMPFC and accG could be more clearly described and discussed in the discussion section; currently, the specific findings and their importance are a bit vague.

We thank the reviewer for the observation which prompted us to expand the discussion of pDMPFC and ACCg results. We now summarise more clearly the results and spell out our interpretation of their importance in the discussion.

5) The methods section indicates that the prediction error variables in the fMRI analyses included (i) the KL divergence and (ii) its sign, which was either 1 or -1. I was **unclear on** why the signed prediction error was restricted to 1 and -1, rather than examining a scaled parametric modulator (i.e. KL divergence x the sign).

We thank the reviewer for their point. The sign of the prediction error doesn't contrast cooperation vs competition per se but rather looks at the changes in social orientation by contrasting trials in which cooperation is increased vs trials in which it is decreased compared to the previous trial. We included in our GLM two regressors related to the prediction error, one with the value of the KL divergence (which is by its definition, always positive) representing the surprise associated with the change in the co-player position and one with the direction of the change (encoded through a signed value). Using simultaneously the KL divergence and the signed prediction error produces two

perfectly collinear regressors in half the trials. To test the full parametric effect of the different clusters in TPJ we run two control GLMs (see methods): one, as suggested by the reviewer, including only a signed prediction error (signed KLD) and another one with four unmodulated regressors for four different trial groupings based on the KLD value and its sign. Whilst in the first GLM activations in TPJ remained under threshold, probably due to the fact that the activity encodes prediction errors asymmetrically across the two contexts, in the second control GLM we see a clear modulation of raTPJ activity with value of prediction errors signalling increases in competition of the co-player (see suppl. Figure X) whilst activations in rpTPJ show a u-shaped relationship with the value of KLD, consistent with our interpretation of a signal reflecting the sign or the absolute value of the prediction error, respectively. We now comment on this additional analysis in the main text (lines 454-460).

6) Can the authors comment on how well subjects understood the extent to which different choices would be cooperative or competitive?

We believe players had a good level of understanding of the implications of different choices. The instructions of the task included examples of different outcome distributions for different positions of players and target that emphasised the trade off between probability of winning the trial individually (competition) vs probability to win more as a couple (cooperation). Furthermore players had a chance to familiarise with the set up in a behavioural session ahead of the fMRI session. At the end of the behavioural session they were asked questions about their strategy which revealed a good understanding of the social implication of their choices, namely that they described or showed to understand their behaviour as cooperative in the cooperative condition and conversely in the competitive condition. We have their response in the new Supplementary Table 3.

Minor points:

1) In describing the three factors shaping cooperation in the introduction, the third factor (“how dyads interact with each other”) can be broadened to an array of social consequences, which can include both reciprocity in repeated interactions (Dal Bo, Dal Bo & Frechette, 2011) and reputation (Barclay & Willer, 2007; Feinberg, Willer, & Schultz, 2014; see Kraft-Todd et al., 2013, for further discussion)

We thank the reviewer for pointing us towards this literature. We have now added these references in the point suggested.

Reviewer #3 (Remarks to the Author):

This manuscript describes the results of an fMRI study in which participants engage in an extended form of the prisoner's dilemma game aimed at exploring neural computations underlying the capacity to co-operate or compete with an opponent. The authors describe behavioral evidence that participants are sensitive to contextual manipulations that change the benefit of competing, in that they choose to compete less in contexts designed to promote cooperation and more in contexts designed to promote competition. A Bayesian model is used to account for participants trial-by-trial behavior. Various models are compared with different levels of complexity, ranging from merely copying the opponent's position, adding in a bias toward cooperation, incorporating mechanisms for tit for tat and so on. When testing for updates about predictions concerning the expected level of cooperation, activity was reported in posterior STS correlating with the ML divergence of the posterior and prior, describe the amount of update occurring in the predictions – both a signed and unsigned version of this signal was found in separate areas of STS. Furthermore, activity in mPFC correlated with other components of the model, and in which activity was particularly dependent on changes in the social context.

By using a novel task to examine cooperative/competitive decisions the authors have generalised these findings to a more continuous setting as well as replicated previous results about a contribution of TPJ in encoding social prediction errors. The report of different areas of TPJ correlating with signed and unsigned prediction errors adds further insight to the functional properties of this area in social prediction error coding. However, the overall sense from this manuscript is that the findings do not necessarily represent a major advance in our understanding of how the brain mediates decisions about cooperating or competing beyond what is already known about the role of TPJ and dmPFC in social computations. A number of additional control conditions would seem to be needed to make a claim that the authors have identified specific evidence for the neural correlates of cooperation/competition. Furthermore, it is not clear that the authors have clear evidence for Bayesian inference above and beyond other possible model implementations. In addition, a number of concerns are noted about the modelling – which can potentially be remedied with additional analyses.

We thank the reviewer for their remarks and suggestions which we have done our best to tackle.

(1) It is not clear from this manuscript that the authors' reported findings are specifically related to co-operation and competition decisions as opposed to corresponding to social decision-making more generally. The signal the authors report in pSTS corresponds to an update related to a difference between expected and observed behavior of the other agent. In the model and task used, this is couched in terms of cooperation vs competition, but such a signal might not be specific to such a circumstance at all, but simply to any situation where an agent needs to make a prediction about another's behavior and update that prediction. Although differences in signal in mPFC were reported as a function of different contexts, such a manipulation changed the riskiness and potential cost of any decision beyond any specific effects on the participant's likelihood to cooperate vs compete. The addition of control conditions such as a non-social condition matched for the gain/loss profile of a decision, but without a social element, or a control condition with a social element but no cooperative/competitive component could help address some of those concerns about specificity.

We thank the reviewer for their observation which prompted us to clarify our stance. We did not intend to claim that the neural activity we report is reflective of areas which are uniquely or specifically involved in the processing of the cooperation-competition trade-off. Assessing specificity for this, or for any social process, is complex and requires a different approach to that used in this experiment (Lockwood, Apps & Chang, 2020, TICS). In fact, we agree it is entirely possible that there are no areas that are exclusively or specifically involved in cooperative or competitive behaviour, and thus it wasn't our intention to suggest that the aim of this paper was to try and identify cooperation/competition specific neural signals.

The aim of this manuscript was to examine how people continuously shift their behaviour when interacting with another, in a task where shifting their behaviour indicated the desire to more strongly cooperate or compete. Notably activity was identified in regions that have been implicated in processing information in a Bayesian manner and in other social processes. Thus, our main conclusions centre around how these regions, linked to social information processing previously, may also guide

decisions about how cooperative we wish to be, by tracking the social context and information about the actions of others. However, we appreciate how the previous version of the manuscript may have left these points unclear. Accordingly we have toned down statements in the text that might have suggested that TPJ or mPFC can be thought of as the seed of cooperation/competition trade-off and we added a sentence to the discussions that defines the scope of the inference that we believe our data can afford (lines 737-744).

On the point of additional control conditions, we think it would be difficult to define them in a way that maintains the gain/loss profile dependency on another “non social” agent’s behaviour, without such agent still being perceived as social. Indeed under a common currency framework it would be safe to assume that mPFC would respond to both social and non-social economic games, but we are primarily focused on understanding the extent to which the activity of this area is additionally *modulated* by the social context and reflects elements suggestive of encoding the cooperation-competition trade off.

We can however provide some anecdotal evidence that participants understood the cooperative/competitive nature of the task from their response to a post-experiment questionnaire, a sample of which we now report in Supplementary Table 3 in the supplementary material. Furthermore, we now include in the paper a number of models that are essentially non-social as they do not consider or infer anything about the hidden intentions of the other player. The updated modelling evidence provides further justification for the idea that the behaviour is social in nature as it is better explained by “social” models than through non-social alternatives.

(2) The authors make a strong case for the Bayesian model that they use – essentially arguing that their data provides evidence that humans use a Bayesian strategy (or Bayesian principles as they put it) to solve the cooperation/competition problem. The word Bayesian is echoed in the title. However, the authors do not compare their Bayesian strategy to a non-Bayesian one, thus we cannot ascertain whether or not a Bayesian strategy is necessarily the best way to explain participants behavior. At the simplest, one could also imagine some sort of RL mechanism in which current behavior is adjusted based on past feedback (whether one won or not on the previous trial for pursuing a particular option). One could also imagine more sophisticated non-Bayesian mechanisms also tracking the opponent’s strategies and implementing a choice accordingly. There is nothing in principle wrong with the Bayesian modeling approach the authors used– it is convenient and often useful to deploy such a modeling approach– but it is a strong claim for the authors to contend that they have evidence participants specifically use a Bayesian strategy and that they necessarily reveal neural correlates thereof – when other non-Bayesian models could potentially explain the results, especially if constructed to perform some form of social inference/learning about others’ actions akin to that which is performed by their Bayesian model.

We thank the reviewer for their comments that prompted us to significantly expand our modelling work in two directions.

First we developed a number of purely reactive, non-Bayesian models (“simple” models 1-2-3-4 in the new figure.3) which assume that players only reciprocate either the last

position of their counterpart, their last change in position, or a combination of the two. All these models are significantly outperformed by the Bayesian models that fit choices based on expectations of their counterparts' positions. Second, to confirm that players were trying to predict others' positions rather than just reciprocating preceding choices, we ran regression models to explain participants' choices based on both the last position of the co-player and its Bayesian expectation in the following trial. We found that expected positions are significantly better predictors than preceding choices (see new supplementary figure 6b).

Both these pieces of evidence point to the fact that whilst players implement tit-x-tat, they do so in a way that considers all past behaviour of their co-player, effectively discounting their latest choice with prior decisions. We believe it is not surprising that, after a number of trials, players expect their co-player to behave in a certain way and we think that this can allow their behaviour to be more resilient. For example, if their counterpart deviates suddenly in a more competitive direction from their usual cooperative position, Bayesian participants would discount this sudden change if there was a consistent history of cooperation, thus making their cooperation more robust to single, potentially accidental, deviations. A Bayesian expectation naturally captures this intuitive behaviour. We have now added a paragraph (lines 344-352) to address this point in the text.

Secondly, following the reviewer's suggestion, we developed a number of feedback based models that try to explain choices based on whether a participant has won or lost the previous trial. We explored both the possibility that following a loss, participants would become more competitive, as one could expect they would, to increase their chance to win, as well as the notion that, following a loss, they might reverse the direction of their latest tendency (that is, becoming more cooperative if the previous update made them more competitive, or viceversa), whilst following a win they might reinforce it, thus adopting a more classic reinforcement setting. All these non-Bayesian models failed to predict choices better than any of the Bayesian ones, even when we combined them with simple non-Bayesian titXtat strategies (see new figure 3 and the Modelling section in the methods).

All this provides in our view strong evidence that our participants relied on a tit-X-tat strategy founded over a Bayesian expectation of their co-player future choices.

(3) On a related point – the relationship between the model predictions and human behavior is unclear. While plots of model fits are provided for each subject, it is hard from these plots to gauge how well the models are doing in capturing trial by trial behavior. Posterior predictive checks are needed to ascertain to what extent the various models utilized can capture key behavioral patterns in the human choice data. It would also be important to implement a **model simulation parameter recovery analysis** to determine to what extent it is possible to uniquely identify each of the parameters included in their model during the model fitting process. Model confusability analyses would also help to identify which models actually make unique behavioral predictions.

We thank the reviewer for this comment which prompted us to perform a model simulation parameter recovery analysis. Our results (reported in the supplementary

material and in the new supplementary figure 6) suggest that the parameters of our winning model are to a large extent recoverable, with a high value of correlation of the true parameters with the recovered ones for all four parameters of the winning model. We observe a lower correlation and lower degree of recoverability for the tit-x-tat and the social-bias parameter, which is not surprising as we found them to be highly anticorrelated, as thoroughly reported in our manuscript (lines 383 to 393). We don't think this to be a shortcoming of the model but an inherent strategic trade-off between a fixed cooperative strategy and a more reactive titxtat alternative which is also reflected in the strength of the neural representations in the cingulate cortex, whereby we found that the representation of increases of cooperation for self positively correlated with the social bias parameter and negatively correlated with the titXtat parameters for all three cingulate clusters and decreasingly so while moving forward along the rostro-caudal axis (see supplementary figure 5e).

(4) It is rather unclear what the mPFC results actually signify in terms of adding to our understanding about how cooperative strategies are implemented at the computational or neural level. For instance, what is the functional significance of a neural signal in dmPFC that shows different signs of social prediction error as a function of different social contexts? What does this signal tell us about how the brain implements this type of decision-making? The impression is that these findings have emerged from an exploratory analysis and that they are not easily interpretable.

We thank the reviewer for their comment. Whilst we agree that the interpretation is not always straightforward, we do not believe that our findings are exploratory. The novelty of this study lies in the ability to show how neural activity in social areas can be significantly modulated by the social context and elucidate the network of areas involved in arbitrating the cooperation/competition trade off. The neural signal in dmPFC was encoding social prediction errors differently according to the context and was predictive of future change in behaviour. As we speculate in the discussion, these results align with past evidence suggesting that this area is involved in shifting preferences to align to other people's behaviour or social norms, which are likely to be perceived as different in different contexts. However, the evidence available through our study can't rule out competing hypotheses about the specific role of this area. Ultimately, exploring the functional significance of the areas we identify would require carefully calibrated experiments to probe different hypotheses about their roles in different social contexts, something which were beyond the scope of our work. We added a paragraph in the discussion to highlight this as a future direction for new studies employing this or similar paradigms.

REVIEWER COMMENTS

Reviewer #1 (Remarks to the Author):

Thank you for a responsive revision. The following two issues remain:

1) The response to previous comment 2) was not entirely clear. The BIC figure up to model 8 has changed from the initial submission without comment by the authors. The response does not address the original issue about model selection, which concerned the BIC of models 7 and 8 being lower than the BIC of model 6 (not just the difference being small) in the previous figure. Given that the BIC value of the best model in the caption of Figure 3 (BIC = 5556) in the previous manuscript doesn't match the y value in the plot (below 5000), the BIC value of the current figure seems to be more reasonable, but I don't find any explanations about why the BIC value of the best model has changed (previously 5556, currently 5553), the difference between BIC correct (previous) and BIC (current), and why the BIC value in the plot has changed in the revision.

2) The supplementary table seems to be missing.

Reviewer #2 (Remarks to the Author):

I think the authors have done a nice job addressing the concerns I raised in the original manuscript, including clarifying some analyses/findings and addressing links to prior work. Accordingly, I think this manuscript can make a nice contribution to the literature on human cooperation by bringing together multiple components of cooperation together in behavior and the brain.

Reviewer #3 (Remarks to the Author):

I appreciate the authors work on addressing the reviewer comments. However, I still have several outstanding issues:

Regarding my original point 2, while the expanded model comparison and additional analyses that the authors have implemented does help to rule out simple strategies – e.g. just tracking the last behavior of the other agent or changing strategy based on reward feedback, I'm still not persuaded that the authors have definitive evidence that individuals are necessarily implementing a Bayesian strategy. One could still think of non-Bayesian ways to keep track of the other agents expected behavior that integrates over their past history of choices (as opposed to just the last choice), and models that do this are not tested. One could also think of imperfect Bayesian models such as for examples models with a forgetting parameter etc.

The model they propose potentially provides an approximation of the true strategy that people are using – but we don't have to conclude that people are actually solving things using Bayesian inference. To do so requires a lot more evidence about exactly how the belief updating is happening and evidence that people are actually representing full probability distributions over beliefs (use of KLD as a measure of surprise in the fMRI analysis doesn't provide definitive evidence of this as simpler updating rules based on e.g. just comparing the mean of posterior and prior distributions would likely produce similar activation patterns). I suggest that they simply tone down or condition their claims that they have evidence for Bayesian inference in the title and throughout the manuscript.

As for my original point 3, I cannot find Supplementary Figure 6 in the revised submission, so I cannot evaluate the authors' responses to my concerns in that regard.

Also, as an aside, the resolution of the figures in the pdfs that were provided was very poor making it

very hard to read the figures, but fortunately the original word docs were also available which appears not to have those problems.

REVIEWER COMMENTS

Reviewer #1 (Remarks to the Author):

Thank you for a responsive revision.

We thank the reviewer for their positive assessment. We now address the reviewer's remaining comments below.

The following two issues remain:

1) The response to previous comment 2) was not entirely clear. The BIC figure up to model 8 has changed from the initial submission without comment by the authors. The response does not address the original issue about model selection, which concerned the BIC of models 7 and 8 being lower than the BIC of model 6 (not just the difference being small) in the previous figure. Given that the BIC value of the best model in the caption of Figure 3 (BIC = 5556) in the previous manuscript doesn't match the y value in the plot (below 5000), the BIC value of the current figure seems to be more reasonable, but I don't find any explanations about why the BIC value of the best model has changed (previously 5556, currently 5553), the difference between BIC correct (previous) and BIC (current), and why the BIC value in the plot has changed in the revision.

We thank the reviewer for noticing this discrepancy. The reviewer is absolutely right. There was a mistake in the previous submission figure, not the actual values used for the model comparison. We have therefore corrected the figure in the revised submission. The values included in the revised figure have been double-checked and are now correct. We apologise for the confusion.

2) The supplementary table seems to be missing.

We are really sorry but for some reason the originally revised supplementary information with the new supplementary tables was not attached in the previous submission. We have now rectified this issue.

Reviewer #2 (Remarks to the Author):

I think the authors have done a nice job addressing the concerns I raised in the original manuscript, including clarifying some analyses/findings and addressing links to prior work. Accordingly, I think this manuscript can make a nice contribution to the literature on human cooperation by bringing together multiple components of cooperation together in behavior and the brain.

We thank the reviewer for their kind words of appreciation of our work.

Reviewer #3 (Remarks to the Author):

I appreciate the authors work on addressing the reviewer comments.

We thank the reviewer for their appreciation of our work.

However, I still have several outstanding issues:

Regarding my original point 2, while the expanded model comparison and additional analyses that the authors have implemented does help to rule out simple strategies – e.g. just tracking the last behavior of the other agent or changing strategy based on reward feedback, I'm still not persuaded that the authors have definitive evidence that individuals are necessarily implementing a Bayesian strategy. One could still think of non-Bayesian ways to keep track of the other agents expected behavior that integrates over their past history of choices (as opposed to just the last choice), and models that do this are not tested. One could also think of imperfect Bayesian models such as for examples models with a forgetting parameter etc.

The model they propose potentially provides an approximation of the true strategy that people are using – but we don't have to conclude that people are actually solving things using Bayesian inference. To do so requires a lot more evidence about exactly how the belief updating is happening and evidence that people are actually representing full probability distributions over beliefs (use of KLD as a measure of surprise in the fMRI analysis doesn't provide definitive evidence of this as simpler updating rules based on e.g. just comparing the mean of posterior and prior distributions would likely produce similar activation patterns). I suggest that they simply tone down or condition their claims that they have evidence for Bayesian inference in the title and throughout the manuscript.

We thank the reviewer for raising this point. Whilst we think our control models and analysis provide additional evidence that our participants were behaving as if they were building expectations about their co-player choices, the reviewer is right in that we can't assert with full confidence that the cooperation-competition trade off is mediated by a full Bayesian mechanism. In other words, we can't provide evidence that our participants represented full prior and posterior probability distributions over the game space nor that they didn't approximate them or the Bayesian inference. Following the Reviewer's suggestion we have now changed the manuscript in two ways. First, we have removed the word Bayesian from the title (which now reads "Neural implementation of computational mechanisms underlying the continuous trade-off between cooperation and competition") and the abstract. Second, we have toned down several passages in the text (including a subtitle and a caption) where we either removed the word Bayesian or we slightly reformulated them.

As for my original point 3, I cannot find Supplementary Figure 6 in the revised submission, so I cannot evaluate the authors' responses to my concerns in that regard.

We believe that supplementary figure 6 was included at page 30 of the originally revised manuscript. However the reviewer is right in that for some reason the

supplementary information section, where the new supplementary figure was also copied, was not attached in the resubmission. We apologised for this and we have now rectified this issue by including everything in the latest revision.

Also, as an aside, the resolution of the figures in the pdfs that were provided was very poor making it very hard to read the figures, but fortunately the original word docs were also available which appears not to have those problems.

We thank the reviewer for highlighting this point. We double checked and rectified the resolution of all figures, which we hope will now be satisfactory when exporting into a pdf.

REVIEWERS' COMMENTS

Reviewer #1 (Remarks to the Author):

Thank you for the revision

Reviewer #3 (Remarks to the Author):

The authors have satisfactorily addressed my remaining concerns.